# SURE: Shift-aware, User-adaptive, Risk-controlled REcommendations

## ABSTRACT

Although Sequential Recommender Systems (SRS) have been well developed to capture temporal dynamics in user behavior, they face a critical gap in formal performance guarantees under preference shifts. When preferences change, predictions often become unreliable, undermining user trust and threatening long-term platform success. To address this challenge, we introduce **SURE** (**S**hift-aware, **U**ser-adaptive, **R**isk-controlled **RE**commendations), a dataset- and model-agnostic framework that provides adaptive recommendation sets with formal coverage guarantees while remaining compact under preference shifts. Specifically, SURE (i) ensures validity through a loss-based change-point mechanism that adaptively updates calibration thresholds upon detecting preference shift, (ii) maintains compact recommendation sets by stabilizing predictions with a Hedge-weighted ensemble of bootstrapped experts, preventing validity from degenerating into impractically large outputs, and (iii) guarantees robustness under non-stationarity by deriving finite-sample bounds that ensure the ensemble's expected set size remains close to the best expert while controlling the utility-based risk in recommendation. Extensive experiments across multiple datasets and base models validate the effectiveness of the proposed framework, which aligns with our theoretical analysis.

## 1 INTRODUCTION

Sequential recommendation systems (SRS) learn temporal dependencies across user interaction sequences to forecast future behavior, making them essential for platforms such as e-commerce, streaming, and location-based services (Hussien et al., 2021; Chang et al., 2017; Rohilla et al., 2021). Much research focuses on developing different architectures, e.g., SASRec (Kang & McAuley, 2018), Caser (Tang & Wang, 2018), and FMLP-Rec (Zhou et al., 2022), which are trained on historical interactions and then deployed with fixed parameters. In their canonical offline form, these models are trained on historical interactions and then deployed with static parameters (Farzad & Bamshad, 2018; Chen et al., 2023), capturing temporal dynamics but remain brittle under preference shifts. As a result, predictions may become unreliable. Periodic retraining can be costly and add latency (Shen & Kurshan, 2023; Zhang et al., 2020), which is unacceptable in high-stakes recommender scenarios.

In recent years, some works have sought to mitigate this challenge by incorporating temporal positional encodings (Li et al., 2020) or segmenting user histories through causal variational frameworks (Wang et al., 2023). However, these approaches still assume locally stable environments and cannot fully adapt to abrupt preference shifts. Other works exploit future user interactions as oracle signals during training (Xia et al., 2025). While promising, this strategy depends on information unavailable in real-time prediction. Importantly, none of these methods provides statistical guarantees on performance under evolving user behavior, a critical vulnerability that undermines the trustworthiness of recommender systems.

As a result, we are motivated to propose a fundamentally new approach: a model-agnostic recommendation framework offering rigorous statistical guarantees for performance under non-stationary user preferences. Specifically, our goal is to construct dynamic and compact prediction sets around recommended items that adaptively adjust to evolving user behaviors and guarantee recommendation performance with high confidence (e.g., 95%) over time.

---

The code and implementation details are available at https://anonymous.4open.science/r/SURE_-02D2

While Conformal Prediction (CP) (Vovk et al., 2005; Angelopoulos & Bates, 2021) can offer a principled approach to the above challenge, it cannot be naively applied due to the violation of the exchangeability assumption and distribut shift in SRS. Fortunately, adaptive conformal approaches exist to handle non-exchangeability (Xu & Xie, 2021). However, applying these frameworks to sequential RS task presents unique challenges: 1) The current adaptive conformal prediction methods (Xu et al., 2024) often rely on a fixed-size rolling window to update confidence sets, which, as discussed by Zaffran et al. (2022), can only work well for stationary residuals. Whereas real-world behaviours in sequential RS environments are highly non-stationary. A fixed window size implicitly assumes a constant shift rate, causing delayed adaptation for fast-shifting users and unnecessary fluctuations for stable ones. 2) Secondly, tight and stable $(1 - \alpha)$-marginal coverage is attainable only when each calibration window contains a sufficiently large and representative sample of residuals (Gibbs & Candes, 2021; Zaffran et al., 2022; Angelopoulos et al., 2024). In sequential RS, however, deep sequence models are trained on short and noisy interaction histories, which may result in unstable model scores and heavy-tailed residual distributions. This instability eventually leads to noisy threshold estimation and overly conservative prediction sets (Barber et al., 2021; Gupta et al., 2019). This raises an important question: Can we design an uncertainty-aware prediction framework that adapts to user-specific, non-stationary shift while (1) maintaining compact prediction sets even under high uncertainty in model outputs, and (2) dispensing with a fixed-window hyperparameter yet still guaranteeing $(1 - \alpha)$- marginal coverage with at least $(1 - \delta)$-confidence?

To answer these challenges, we propose **SURE** (**S**hift-aware, **U**ser-adaptive, **R**isk-controlled **RE**commendations), a model-agnostic framework that outputs dynamic recommendation sets adapting to user-specific preferences shifts while offering formal guarantees. Specifically, SURE improves robustness by maintaining an ensemble of base recommenders, each trained on a different bootstrap of user–item interactions with their prediction sets aggregated via Hedge weighting, which, while maintaining validity, automatically favours experts producing compact recommendations. It also employs segmentation-based recalibration that triggers localized threshold updates using a loss-based metric instead of a fixed rolling window. Subsequently, we prove that SURE controls both prediction set size via variance-controlled aggregation and utility-based risk at inference time through adaptive threshold calibration under preference shifts. We illustarte the framework in Figure 1 in *Appendix*.

Our contributions are summarized as follows:

- Firstly, we formulate the sequential recommendation problem from the perspective of an uncertainty-aware prediction task, and propose a reliable and adaptive framework- SURE, which generates compact yet valid prediction sets with user-specified $\alpha$-risk under non-stationary preferences.

- We then develop Dynamically Adaptive Uncertainty-aware Optimization (DAUO), an efficient Hedge-based ensemble optimization algorithm that jointly updates ensemble weights and risk thresholds to balance prediction set compactness and risk coverage, thereby achieving the objectives of SURE.

- Technically, we introduce a scalar loss-based shift metric that combines a relative loss-discrepancy and a concept-sensitive divergence to quantify user preference shift, thereby enabling dynamic segmentation and localized threshold recalibration.

- Theoretically, we establish statistical guarantees for SURE. Specifically, we show that (1) the expected size of the ensemble prediction set never exceeds the best individual model's size at that timestamp, up to a variance-controlled slack (Theorem 5.1); and (2) the expected utility-based risk at inference stays within a provable margin of $\alpha$ with probability at least $1 - \delta$, even under shifting user preferences (Theorem 5.2).

- Empirically, we conduct extensive experiments using diverse recommendation base models and benchmark datasets. We evaluate SURE against preference-aware recommender baselines in terms of recommendation performance and against conformal prediction methods with respect to recommendation set compactness and coverage guarantees. The results, as presented in Section 6, confirm the effectiveness and robustness of SURE, consistent with its theoretical foundations.

## 2 RELATED WORK

### 2.1 SEQUENTIAL RECOMMENDATION SYSTEMS (SRS)

SRS initially modeled item–to–item transitions with Markov chains (Rendle et al., 2010) or factorization approaches Rendle et al. (2009) that accounted for short-range dependencies in user histories. Deep learning models such as GRU4Rec (Hidasi et al., 2015), convolutional architectures, and transformer-based methods (e.g., SASRec (Kang & McAuley, 2018), BERT4Rec (Sun et al., 2019)) extended this to capture long-term dependencies. However, while these models effectively learn temporal dynamics, they often struggle to remain reliable when user preferences shift abruptly (Quadrana et al., 2018; Pan et al., 2024). Recent works have sought to address non-stationarity or preference shifts explicitly by disentangling user preferences through self-supervision (Ma et al., 2020), modeling temporal intervals in self-attention (Li et al., 2020), or separating stable and shifting preferences via causal reasoning (Wang et al., 2023). A parallel line of work focuses on predictive uncertainty in recommender systems. Coscrato & Bridge (2023); Xu et al. (2024) investigate fundamental limits of top-$N$ recommendation accuracy using information-theoretic bounds highlighting the increasing importance of principled uncertainty modeling. Paliwal et al. (2024) propose Predictive Relevance Uncertainty to estimate prediction reliability based on distance to training samples, while Cui et al. (2024) develop a Bayesian deep collaborative filtering model coupled with an uncertainty-aware ranking to improve trustworthiness in online physician recommendations. More recently, variational and stochastic sequence models. Fan et al. (2021); Fang et al. (2020); Wang et al. (2022) have explored uncertainty-aware sequential recommendation. However, these approaches, while powerful, still lack finite-sample guarantees on recommendation quality under evolving preferences.

### 2.2 CONFORMAL PREDICTION

Conformal Prediction (CP) can quantify models' uncertainty and can provide a finite sample guarantee by creating distribution-free prediction sets that contain the true outcome with a user-specified coverage probability (Vovk et al., 2005; Shafer & Vovk, 2008; Romano et al., 2019). Classical CP (Angelopoulos & Bates, 2021) assumes exchangeability and uses a calibration split to choose a global threshold; Some other methods like Inductive CP (Papadopoulos, 2008) consider the full dataset. To remain valid under temporal shift, online (Angelopoulos et al., 2024; Wu et al., 2025) and adaptive CP techniques (Gibbs & Candes, 2021; Zaffran et al., 2022; Xu et al., 2024; Liang et al., 2025) have been developed that update calibration statistics on sliding windows or with an adaptive rate of change in the global threshold. Some works have extended CP to recommender systems. Kagita et al. (2022; 2023) extended top-$N$ recommendation with conformal guarantees. However, these approaches do not account for non-stationarity or change in user preferences in RS.

## 3 PRELIMINARIES

We first introduce the notations used in this paper. We consider $m$ users and n items represented by $\mathcal{U} = \{u_k\}_{k=1}^m$, and $\mathcal{I} = \{i_k\}_{k=1}^n$. For brevity, we use $u$ and $i$ to denote a user and an item in this paper. In a sequential recommendation setting, every user $u$ has a chronological sequence of interacted items, denoted as $\mathcal{H}_u = [i^1, i^2, \ldots, i^{|T_u|}]$ where $i^t \in \mathcal{I}$ represents an item interacted with by user $u$ at time step $t$, and $|T_u|$ denotes the length of the sequence for user $u$. The objective of the SRS is, given the historical interaction sequence $\mathcal{H}_u$ for each user $u$ predict the next item they are likely to interact with. Specifically:

$$i^{t+1} = \arg\max_{i \in \mathcal{I}} \mathcal{M}(i \mid \mathcal{H}_u), \tag{1}$$

where, $\mathcal{M}(i \mid \mathcal{H}_u) : \mathcal{I} \times \mathcal{H}_u \to [0, 1]$, denotes the underlying recommender model.

Given the dynamic nature of user preferences, however, there is no guarantee of the model's performance. This limitation motivates us to explore the creation of dynamic recommendation sets that adapt with changing user preferences, which we discuss next.

## 4 THE PROPOSED FRAMEWORK

In this section, we propose Shift-aware, User-adaptive, Risk-controlled REcommendations (SURE), a novel framework designed to provide compact recommendations that adapt to evolving user

preferences with theoretical performance guarantees in a sequential recommendation setting. We begin by defining the construction of the dynamic prediction set $\mathcal{C}^{t+1} \subseteq \mathcal{I}$ for a single underlying model, which is guided by a timestep-dependent threshold parameter $\lambda^t \in \Lambda \subset \mathbb{R}$. Specifically:

$$\mathcal{C}_{\lambda^t}^{t+1}(\mathcal{H}_u) \;=\; \big\{\, i \in \mathcal{I} \mid \mathcal{M}(i \mid \mathcal{H}_u) \geq \lambda^t \big\}. \tag{2}$$

For brevity we will refer to $\mathcal{C}_{\lambda^t} \;:=\; \mathcal{C}_{\lambda^t}^{t+1}(\mathcal{H}_u)$. Our goal is, given the user-defined error rate $\alpha \in [0, 1]$, for every timestamp, the recommendations created ensure:

$$R(\mathcal{C}_{\lambda^t}) \;\leq\; \alpha. \tag{3}$$

The risk $R(.)$ in Equation (3) is defined as:

$$R(C_{\lambda^t}) = \mathbb{E}_U[\mathcal{L}_u(\mathcal{C}_{\lambda^t})], \tag{4}$$

where $\mathcal{L}_u(.)$ is the bounded user utility-based loss function defined as:

$$\mathcal{L}_u(\mathcal{C}_{\lambda^t}) = 1 \;-\; U_{metric}\big(i_{rel}^{t+1}, \mathcal{C}_{\lambda^t}\big). \tag{5}$$

Here, $U_{metric}(\cdot)$ represents generalized recommendation metric (such as Recall or NDCG) that measures performance of recommendation set $\mathcal{C}_{\lambda^t}$ for any user $u$ given the relevant item $i_{rel}^{t+1}$.

The threshold $\lambda^t$ in Equation (2) is learned from model scores, which are highly sensitive to the quality of the underlying recommender. When models are trained on sparse user histories, as is common in RS Bertin-Mahieux et al. (2011); Cho et al. (2011), the resulting scores can become unstable, often leading to heavy-tailed residual distributions, which in turn destabilize threshold estimation and might result in overly conservative prediction sets.

To address this, we propose using an ensemble of $L$ base models:

$$\mathbb{M} \;=\; \big\{\mathcal{M}^1,\, \mathcal{M}^2,\, \ldots,\, \mathcal{M}^L\big\}, \tag{6}$$

where $L$ is the number of models in the ensemble and each $\mathcal{M}^\ell$ is obtained, for example, by training on a bootstrap sample of the full user set $\mathcal{U}$, i.e., $\mathcal{U}^\ell \subseteq \mathcal{U}$.

Firstly, for each model $\mathcal{M}^\ell$, we generate a prediction set $\mathcal{C}_\ell^{t+1} := \mathcal{C}_{\lambda_\ell^t}^{t+1}(\mathcal{H}_u)$, guided by its own threshold $\lambda_\ell^t$ and ensuring the per-model analogue of Equation (3) is satisfied. Next, we aggregate these sets, generated using $\boldsymbol{\lambda}^t = \{\lambda_\ell^t\}_{\ell=1}^L$, into an ensemble-based recommendation. Specifically:

$$\mathcal{C}_{\boldsymbol{\lambda}^t}^{\mathrm{agg}} \;=\; \mathcal{A}\Big(\{\mathcal{C}_\ell^{t+1}\}_{\ell\,:\,u\in\mathcal{U}^\ell},\, \mathbf{w}^t\Big), \tag{7}$$

where $\mathcal{A}(\cdot, \mathbf{w}^t)$ is an aggregation operator that merges the individual set predictors $\mathcal{C}_\ell^{t+1}$ using a weight distribution $\mathbf{w}^t \in \Delta^L$, the $(L-1)$-dimensional probability simplex (i.e., $\Delta^L = \{w \in \mathbb{R}^L : w_\ell^t \geq 0,\ \sum_{\ell=1}^L w_\ell^t = 1\}$), where each $w_\ell^t$ determines the contribution of model $\mathcal{M}^\ell$ at time $t$.

The aggregation operator $\mathcal{A}(\cdot, \mathbf{w}^t)$, following Gasparin & Ramdas (2024), is defined as:

$$\mathcal{A}\Big(\{\mathcal{C}_\ell^{t+1}\}_{\ell=1}^L,\, \mathbf{w}^t\Big) = \left\{ i \in \mathcal{I} \ \middle|\ \sum_{\ell=1}^L w_\ell^t \cdot \mathbf{1}\big(i \in \mathcal{C}_\ell^{t+1}\big) > \frac{1 + k(t)}{2} \right\}. \tag{8}$$

Items $i$ are included in the ensemble set if their total weighted support across base models exceeds the randomized threshold $\frac{1+k(t)}{2}$, where $k(t) \sim \mathrm{Uniform}[0, 1]$ introduces mild stochasticity to discourage marginal inclusions. To favor models that produce efficient sets, we follow Freund & Schapire (1997) and adaptively update the weights $\{\mathbf{w}^t\}_{t=1}^T$ based on the cardinality of the prediction sets produced.

Specifically, let $s_\ell^t$ denote the cardinality of the prediction set produced by base model $\mathcal{M}^\ell$ at time $t$, i.e., $s_\ell^t = |\mathcal{C}_\ell^{t+1}|$, and let the cumulative size up to time $t$ be $S_\ell^t = \sum_{\tau=1}^t s_\ell^\tau$. Then, for a learning rate $\eta \geq 0$, we update the weights as:

$$w_\ell^{t+1} = \frac{\exp(-\eta S_\ell^t)}{\sum_{j=1}^L \exp(-\eta S_j^t)}, \qquad \text{with } \mathbf{w}^1 = \big(\tfrac{1}{L}, \ldots, \tfrac{1}{L}\big). \tag{9}$$

Meanwhile, another tough challenge to tackle is the evolving user preferences. As user preferences change, the model's threshold $\lambda_\ell^t$ learned over previous timestamps may fail to ensure Equation (3). Hence, to maintain statistical validity and capture changes in user preferences, we introduce loss-based shift metrics. For each base model $\mathcal{M}^\ell$, we quantify preference change via the loss discrepancy distance ($d_\ell^{\mathrm{ldd}}$) and the concept-sensitive divergence ($d_\ell^{\mathrm{con}}$) respectively.

To define Loss Discrepancy Distance (LDD), we draw inspiration from the $\mathcal{H}\Delta\mathcal{H}$ divergence definition in Ben-David et al. (2010) by replacing its binary–disagreement indicator with a generalized bounded predictive loss to measure the maximum discrepancy between a reference model and other models across timepoints $t$ and $t' < t$. Specifically:

$$d_\ell^{\mathrm{ldd}}(t, t') = \max_{\mathcal{M}' \in \mathbb{M}, \, \mathcal{M}' \neq \mathcal{M}^\ell} \left| \log \left( \left| \frac{L_t(\mathcal{M}^\ell) - L_t(\mathcal{M}')}{L_{t'}(\mathcal{M}^\ell) - L_{t'}(\mathcal{M}') + \epsilon} \right| + \epsilon \right) \right|, \tag{10}$$

where $L_t(\mathcal{M}^\ell)$ denotes a generalized loss function (e.g. cross entropy) of model $\mathcal{M}^\ell$ at time $t$, and $\epsilon > 0$ ensures stability. Similarly, to capture concept-sensitive divergence, we define a hazard-style term that compares the model's loss across $t$ and $t'$ to the sum of the least individual losses. Formally:

$$d_\ell^{\mathrm{con}}(t, t') = \log \left( \frac{L_t(\mathcal{M}^\ell) + L_{t'}(\mathcal{M}^\ell) + \epsilon}{\min_{\mathcal{M} \in \mathbb{M}}(L_t(\mathcal{M})) + \min_{\mathcal{M} \in \mathbb{M}}(L_{t'}(\mathcal{M})) + \epsilon} \right). \tag{11}$$

We then combine the relative loss-discrepancy and the concept-sensitive divergence into a single scalar loss-based shift metric of preference change:

$$d_\ell^{\mathrm{pref}}(t, t') = d_\ell^{\mathrm{ldd}}(t, t') + d_\ell^{\mathrm{con}}(t, t'). \tag{12}$$

To localize preference shifts, we embed $d_\ell^{\mathrm{pref}}$ in a Bayesian change-point model. At each timestamp $t$, we place a posterior probability distribution $p_\ell(\cdot)$ over candidate segment starts $c_\ell^t \in \{ c_\ell^{t-1} + k \mid k = 0, \ldots, t - c_\ell^{t-1} \}$ as follows:

$$p_\ell\big(c_\ell^t = c_\ell^{t-1} + k \mid t\big) = \frac{\exp\big[-\beta \, d_\ell^{\mathrm{pref}}(c_\ell^{t-1} + k, \, t)\big] \, (t - c_\ell^{t-1} - k + 1)^\gamma}{\displaystyle\sum_{j=0}^{t-c_\ell^{t-1}} \exp\big[-\beta \, d_\ell^{\mathrm{pref}}(c_\ell^{t-1} + j, \, t)\big] \, (t - c_\ell^{t-1} - j + 1)^\gamma}, \tag{13}$$

where $\beta > 0$ tunes shift sensitivity and $\gamma \geq 0$ controls the segment-length bias.

We pick the segment boundary for each model i.e. $c_\ell^t = c_\ell^{t-1} + \arg\max_k p_\ell\big(c_\ell^{t-1} + k \mid t\big)$, set the stable window $\mathcal{W}_\ell^t := [\, c_\ell^t, \, t\,]$ and then calculate the average risk of the window as follows:

$$\bar{R}_\ell^t = \frac{1}{|\mathcal{W}_\ell^t|} \sum_{\tau \in \mathcal{W}_\ell^t} R(\mathcal{C}_\ell^\tau). \tag{14}$$

Finally, to adaptively maintain statistical guarantees under detected user preference shift, calibration threshold is updated as follows:

$$\lambda_\ell^{t+1} = \lambda_\ell^t - \rho\big(\bar{R}_\ell^t - \alpha\big), \tag{15}$$

where $\rho > 0$ is a step size. The threshold $\lambda_\ell^{t+1}$ decreases when segment risk exceeds $\alpha$, expanding the prediction set to restore validity, and increases when risk falls below $\alpha$, thus achieving automatic recalibration.

To this end, we complete modeling of proposed framework. To output user-wise dynamic prediction sets, we instantiate it through DAUO ( Dynamically Adaptive Uncertainty-aware Optimization) algorithm to learn parameters $\lambda_l^t$ and weight vector $\mathbf{w}^t$. Algorithm is in Section A.3 *Appendix*.

**Prediction Set Construction:** At every interaction, the DAUO algorithm considers two adaptive parameters: the current calibration threshold $\lambda_\ell^t$ and the ensemble weight vector $\mathbf{w}^t$. When a user $u$ with history $S_u$ arrives, the algorithm first evaluates every base model $\mathcal{M}^\ell$ to obtain the individual prediction sets (Equation (2)). It then combines these sets through the weighted majority operator in Equation (8), producing the aggregated recommendation. Since $\lambda_\ell^t$ is updated adaptively to enforce Equation (3) and the Hedge weights are penalized by set size, the resulting prediction set is not only valid, i.e., controls risk at level $\alpha$, but also simultaneously compact.

## 5 THEORETICAL ANALYSIS

In the previous sections, we demonstrate how the DAUO algorithm dynamically learns the threshold $\lambda_\ell^t$ for an ensemble of trained models $\mathcal{M}^\ell$ and updates it via empirical risk estimates over adaptive windows, however, it remains to be seen whether this online calibration guarantees efficient and valid predictions. In this section, we provide a theoretical analysis on (1) the provable upper bound on the ensemble prediction set produced via weighted majority voting, and (2) the threshold $\lambda_\ell^t$, learned from historical user interactions and estimated segmental risk $\bar{R}_t^\ell$, ensures that the true expected risk remains close to the desired threshold $\alpha$ with high probability $1 - \delta$.

**Theorem 5.1** (Expected Aggregator Size). *Let $\mathcal{C}_{\lambda^t}^\ell \subseteq \mathcal{I}$ denote the prediction set produced by base model $\mathcal{M}^\ell$ at time $t$, and let $s_\ell^t := |\mathcal{C}_{\lambda^t}^\ell|$. Let $\boldsymbol{\lambda}^t = (\lambda_1^t, \ldots, \lambda_L^t)$ denote the per-model thresholds such that the ensemble set $\mathcal{C}_{\boldsymbol{\lambda}^t}^{\mathrm{agg}}$ is formed by the randomized weighted majority rule $k(t) \sim \mathrm{Uniform}[0, 1]$ with Hedge weights $\boldsymbol{w}^t \in \Delta^L$. Assuming $\ell^* := \arg\min_\ell s_\ell^t$ is the best expert at round $t$, the expected size of the aggregated prediction set at time $t + 1$ satisfies:*

$$\mathbf{E}_{k(t)}\left[\left|\mathcal{C}_{\boldsymbol{\lambda}^t}^{\mathrm{agg}}\right|\right] \leq s_{\ell^*}^t + \sqrt{2 \ln L \; v_t} + \frac{2}{3} \ln L, \tag{16}$$

*where $v_t := \mathrm{Var}_{\ell \sim \boldsymbol{w}^t}\left(s_\ell^t / |\mathcal{I}|\right) \in [0, 1]$ is variance of normalized set sizes under Hedge distribution.*

*Proof.* Proof with Lemma A.4.1 can be found in Section A.4.1 in *Appendix*. $\square$

**Remark 1.** *Theorem 5.1 shows that expected size of ensemble prediction set is no worse than that of best base model at $t$, up to a variance-dependent slack. As base predictors begin to agree on coverage, the variance $v_t$ diminishes, and the ensemble size approaches the best-case performance.*

**Theorem 5.2** (Expected Risk Control under User Preference Shifts). *Let the DAUO algorithm run over a horizon of length $T$. Assume the Bayesian change-point detector raises $N_T$ preference shifts and let $d_j$ be the detection delay of the $j$-th shift so that $D_T := \sum_{j=1}^{N_T} d_j$. Let $\boldsymbol{\lambda}^T = (\lambda_1^T, \ldots, \lambda_L^T)$ denote the vector of per-model thresholds after round $T$, and let $\mathcal{C}_{\boldsymbol{\lambda}^T}^{\mathrm{agg}}$ denote the ensemble prediction set formed with those thresholds. Let $\mathcal{L}_u(\mathcal{C}_{\boldsymbol{\lambda}^T}^{\mathrm{agg}})$ be the utility-based loss of user $u$ under that ensemble. Given a user batch of size $|\mathcal{U}|$ and a user-defined risk level $\alpha$, then with probability at least $1 - \delta$, the expected utility-based loss at time $T + 1$, using the final threshold $\boldsymbol{\lambda}^T$, satisfies:*

$$\mathbb{E}_{u \sim \mathcal{U}}\left[\mathcal{L}_u(\mathcal{C}_{\boldsymbol{\lambda}^T}^{\mathrm{agg}})\right] \leq \alpha + 2\sqrt{\frac{\log(4|\mathcal{U}|)}{2|\mathcal{U}|}} + \frac{D_T + 2\log(1/\delta)}{T}. \tag{17}$$

*Proof.* Proof with Lemmas A.4.2 to A.4.4 can be found in Section A.4.2 in *Appendix*. $\square$

**Remark 2.** *Theorem 5.2 ensures calibrated $\lambda^T$ guarantees expected risk at time $T + 1$ remains close to user-defined target $\alpha$, with confidence. The bound captures both calibration uncertainty (which decays with user batch size $|\mathcal{U}|$) and change-adaptation error (which vanishes as cumulative delay $D_T$ becomes sublinear in $T$). As both calibration and adaptation improve with scale, expected loss at prediction time $T + 1$ converges to $\alpha$, ensuring reliability even under non-stationary user preferences.*

To sum up, the results establish that our framework, by adaptively calibrating threshold $\lambda_\ell^t$ and leveraging ensemble voting, guarantees control of both recommendation set size and utility-based risk. Specifically, the set size remains competitive with best individual model (up to ensemble variance), and expected loss at time $T+1$ is provably bounded around the user-specified threshold $\alpha$.

## 6 EXPERIMENTS

In this section, we conduct experiments to evaluate the effectiveness of the proposed SURE framework. Specifically, we design experiments to **(1)** validate whether the framework can achieve superior performance in terms of recommendation metrics, i.e., Recall, NDCG and MRR when compared to base models as well as preference-aware baselines, and **(2)** compare performance of the framework with various static and adaptive conformal frameworks in terms of compactness of recommendation set sizes and validity of coverage guarantees **(3)** analyze time efficiency of the proposed SURE framework, **(4)** analyze the influence of hyperparameters, including key conformal parameters $(\alpha, \delta)$ as well as change-point detector settings $(\beta, \gamma)$ and ensemble size $(L)$ on the framework's performance (Section A.7.3 in *Appendix*), **(5)** conduct an ablation study to disentangle the contributions of components in the shift detector (Section A.7.4 in *Appendix*).

## 6.1 DATASETS AND BASELINE MODELS

We conduct experiments on five publicly available datasets across diverse domains: (1) Book-Crossing (book reviews) (Ziegler et al., 2005), (2) Last.fm (music streaming) (Bertin-Mahieux et al., 2011), (3) Taobao (e-commerce) (Jingwei et al., 2020), (4) MovieLens (movie ratings) (Harper & Konstan, 2015), and (5) Gowalla (location-based social network) (Cho et al., 2011). We implement SURE on four base recommendation models selected to represent diverse modeling paradigms: (1) NeuMF (He et al., 2017) (generalized matrix factorization and MLP hybrid), (2) CASER (Tang & Wang, 2018) (convolutional sequence embedding), (3) SASRec (Kang & McAuley, 2018) (self-attention-based sequential modeling), and (4) FMLP-Rec (Zhou et al., 2022) (filter-enhanced feed-forward MLP-based model). For evaluation, we consider both standard recommendation metrics, i.e., Recall, MRR, and NDCG, as well as uncertainty-aware objectives, including coverage guarantees and prediction set size (compactness). On the recommendation metrics, we compare SURE against three preference-aware recommendation models: (1) TiSASRec (Li et al., 2020), (2) CDR (Wang et al., 2023), and (3) Oracle4Rec (Xia et al., 2025). For uncertainty-aware evaluation, we compare against three conformal prediction methods: (1) standard Split Conformal (Vovk et al., 2005), where the threshold parameter $\lambda$ remains fixed; (2) EnbPI (Xu & Xie, 2021), an ensemble estimator with fixed-window calibration; and (3) Online Conformal (Angelopoulos et al., 2024), which uses decaying update rule for threshold $\lambda^t$. Full implementation details and description of datasets, base models, and preference-aware & conformal baselines for reproducibility are provided in Sections A.5 and A.6 in *Appendix*.

## 6.2 EXPERIMENTAL RESULTS

### 6.2.1 RESULTS COMPARED WITH BASE MODELS AND PREFERENCE-AWARE BASELINES

We evaluate SURE framework using four recommendation base models and against three user-preference-aware baselines in terms of standard metrics (MRR, Recall, NDCG). To reflect practical screen/latency constraints, the maximum recommendation set size is capped at 25 items per user. For each backbone model and metric, we define a *Model Ceiling@25* score as the maximum achievable value under its own ranking when limited to 25 items, computed per-user (by taking the shortest prefix containing the relevant item) and then averaged across users. Following prior conformal prediction literature (Angelopoulos & Bates, 2021; Bates et al., 2021; Vovk et al., 2005), we set the error rate $\alpha = 0.05$ and confidence level $\delta = 0.05$, and aim to construct recommendation sets whose realized metrics remain within $\alpha$ of their corresponding Model Ceiling@25 with probability at least $1 - \delta$. To ensure fair comparison, all baselines are evaluated at the same average set size produced by SURE. Results for BookCrossing and Last.fm are reported in Table 1, with additional results for MovieLens, Gowalla, and Taobao in Section A.7.1 in *Appendix*. These results lead to following key observations:

- The proposed SURE framework controls risk within the predefined threshold $\alpha = 0.05$ with high confidence and achieves performance close to the model-specific ceiling across all base models. Consequently, it consistently outperforms all baselines on standard metrics (MRR, Recall, NDCG) across datasets.

- The performance also depends on the base model. For example, the state-of-the-art sequential model FMLP-Rec + SURE consistently outperforms NeuMF + SURE by at least $> 12\%$ on every metric for Book-Crossing and $15\%$ on Last.fm datasets, underscoring the importance of a strong baseline.

- The average set size learned by our SURE framework improves performance of baselines and narrows the gap to their model-specific ceilings, as seen with FMLP-Rec model on Last.fm dataset. However, they still underperform compared to SURE, since a single global prediction size cannot be personalized to individual user satisfaction Kweon et al. (2024).

- While the user-preference aware models generally perform well compared to the baselines across both the datasets, their reliance on temporal cues Li et al. (2020), cross-domain transfer Wang et al. (2023), or future-interaction signals Xia et al. (2025) breaks down under sparsity, domain shift, or real-time constraints. Our uncertainty-aware, shift-adaptive framework doesn't make any such assumption and stays robust in every condition.

- Overall, the results demonstrate the data- and model-agnostic nature of SURE, achieving superior performance across all metrics, models, and datasets.

Table 1: Performance comparisons with base models ( NeuMF, CASER, SASRec and FMLP-Rec ) and user preference aware baselines ( TiSASRec, CDR and Oracle4Rec ) on **Book-Crossing and Last.fM Datasets** using metrics ( MRR, Recall, NDCG ). For SURE, $\alpha$ and $\delta$ are set empirically as 0.05, respectively. Bold indicates the best result, and underline indicates the second best.

| Method | Book-Crossing | | | Last.fm | | |
|---|---|---|---|---|---|---|
| | MRR ↑ | Recall ↑ | NDCG ↑ | MRR ↑ | Recall ↑ | NDCG ↑ |
| Model Ceiling@25(NeuMF) | 0.322 | 0.603 | 0.329 | 0.379 | 0.751 | 0.393 |
| NeuMF | 0.246 | 0.502 | 0.276 | 0.306 | 0.685 | 0.335 |
| NeuMF + SURE (Ours) | 0.289 | 0.557 | 0.302 | 0.336 | 0.701 | 0.354 |
| Model Ceiling@25(CASER) | 0.369 | 0.631 | 0.373 | 0.412 | 0.803 | 0.434 |
| CASER | 0.294 | 0.568 | 0.302 | 0.345 | 0.745 | 0.367 |
| CASER + SURE (Ours) | 0.322 | 0.588 | 0.323 | 0.378 | 0.758 | 0.385 |
| Model Ceiling@25(SASRec) | 0.379 | 0.657 | 0.381 | 0.439 | 0.845 | 0.453 |
| SASRec | 0.327 | 0.556 | 0.329 | 0.369 | 0.766 | 0.389 |
| SASRec + SURE (Ours) | 0.341 | 0.608 | 0.355 | 0.392 | 0.799 | 0.422 |
| Model Ceiling@25(FMLP-Rec) | 0.381 | 0.673 | 0.392 | 0.453 | 0.869 | 0.475 |
| FMLP-Rec | 0.335 | 0.599 | 0.352 | 0.386 | 0.796 | 0.412 |
| FMLP-Rec + SURE (Ours) | **0.357** | **0.628** | **0.368** | **0.402** | **0.812** | **0.432** |
| User Preference-Aware Models | | | | | | |
| TiSASRec | 0.334 | 0.583 | 0.345 | 0.374 | 0.778 | 0.402 |
| CDR | 0.340 | 0.563 | 0.350 | 0.371 | 0.782 | 0.376 |
| Oracle4Rec | 0.345 | 0.603 | 0.353 | 0.390 | 0.798 | 0.422 |

### 6.2.2 RESULTS COMPARED TO CONFORMAL BASELINES

Next, we compare our method with conformal baselines in terms of coverage and set Size. We set error rate $\alpha = 0.10$ and compare on the base recommender models: (1) NeuMF (He et al., 2017), (2) CASER (Tang & Wang, 2018), (3) SASRec (Kang & McAuley, 2018) and (4) FMLP-Rec (Zhou et al., 2022) against different conformal baselines i.e. (1) standard Split Conformal (Vovk et al., 2005), (2) EnbPI (Xu & Xie, 2021), and (3) Online Conformal (Angelopoulos et al., 2024) at next interaction. Each conformal baseline can be interpreted as an ablation of SURE: *Split Conformal* freezes calibration threshold $\lambda$ learned and therefore omits online update in Equation 15. *EnbPI* replaces our Bayesian change-point module with a fixed sliding window, ignoring distributional shifts and the dynamic segmentation of Equation 13. Whereas *Online Conformal* updates $\lambda^t$ at every step using only most recent interaction, thereby discarding historical risk information that our cumulative segment risk in Equation 14 utilizes. Table 2 depicts results on the Book-Crossing dataset, with remaining results present in Section A.7.2 in *Appendix*. They lead to the following observations:

- Our SURE framework achieves the best coverage–size compactness balance. It achieves the required coverage and ensures compact average set size on every base model, underscoring its plug-and-play applicability.

- Split conformal provides compact recommendation sets, but the prediction sets are invalid as the coverage value is around 0.82–0.83, well below the nominal 0.90, thereby revealing the under-calibration under users' preference shifts.

- EnbPI boosts coverage by ~0.03-0.04 compared to split conformal, but does so at the expense of increased prediction set sizes. It highlights the importance of our Bayesian change point detection module to detect the preference shift point.

Table 2: Comparison in terms in terms of coverage and average prediction set size with conformal baselines (Split Conformal, EnbPI and Online Conformal) evaluated on four base recommenders (NeuMF, CASER, SASRec, and FMLP-Rec) using the **Book-Crossing** dataset. The error rate is set as $\alpha = 0.10$. Bold indicates the best result, underline indicates the second best.

| Base Model | Coverage ↑ | | | | Set Size ↓ | | | |
|---|---|---|---|---|---|---|---|---|
| | Split | EnbPI | Online | SURE (Ours) | Split | EnbPI | Online | SURE (Ours) |
| NeuMF | 0.821 | 0.849 | 0.875 | 0.901 | 44 | 43 | 46 | 46 |
| CASER | 0.826 | 0.858 | 0.879 | 0.902 | 44 | 45 | 45 | 44 |
| SASRec | 0.835 | 0.867 | 0.898 | 0.908 | 43 | 47 | 44 | 43 |
| FMLP-Rec | 0.835 | 0.873 | 0.901 | **0.910** | 43 | 47 | 45 | **42** |

- Online conformal narrows the gap as coverage climbs to 0.87–0.90, but remains less efficient than SURE as average set size still exceeds SURE by $1-2$ items. It highlights that on-the-fly calibration alone is susceptible to fluctuations, leading to conservative prediction sets.

- Overall, results demonstrate SURE consistently ensures the best coverage–efficiency trade-off on every baseline model that can ensure valid recommendation sets.

### 6.2.3 TIME EFFICIENCY ANALYSIS

We analyse the computational overhead introduced by SURE on top of the four backbone recommenders (NeuMF, CASER, SASRec, FMLP-Rec). All runs use a single NVIDIA A40 with batch size 256, and each model is trained on 100 epochs. From Table 3, we observe across all five datasets and four baselines, SURE adds at most 1.5 min of wall-clock time. This efficiency occurs because the calibration loop is a single forward pass with simple threshold and change-point updates, with no retraining of network weights. Consequently, the modest extra minute is negligible compared with the performance gain we reported earlier in Table 1. These results confirm that SURE is equivalently efficient and can be scaled to real-world applications.

Table 3: Total time (in minutes) required to train backbone models on five datasets, w and w/o addition of SURE. The "w/ SURE" setting includes backbone training plus 50-step calibration. The calibration parameters $\alpha$ and $\delta$ are both set to 0.05.

| Model | Training | Datasets | | | | |
|---|---|---|---|---|---|---|
| | | Book-Crossing | Taobao | Last.fm | MovieLens-1M | Gowalla |
| NeuMF | w/o SURE | 28.3 | 40.2 | 18.5 | 15.2 | 20.3 |
| | w/ SURE | 29.5 | 41.6 | 19.9 | 16.4 | 21.4 |
| CASER | w/o SURE | 42.3 | 60.4 | 29.5 | 25.9 | 32.6 |
| | w/ SURE | 43.8 | 61.8 | 30.9 | 27.4 | 33.9 |
| SASRec | w/o SURE | 35.2 | 47.1 | 24.4 | 19.1 | 25.3 |
| | w/ SURE | 36.5 | 48.4 | 25.8 | 20.4 | 26.7 |
| FMLP-Rec | w/o SURE | 31.6 | 44.9 | 23.0 | 17.8 | 23.6 |
| | w/ SURE | 32.9 | 46.4 | 24.5 | 19.1 | 25.0 |

## 7 CONCLUSION

This paper address important problem of evolving user preferences that undermine reliability of SRS. To address it, it presents SURE framework, which generates user-specific, dynamic recommendations that evolve with preference shift, guaranteeing performance while keeping them compact. SURE is dataset and model agnostic and we validate its effectiveness through theoretical analysis and extensive empirical studies. Since thresholds and ensemble weights are updated externally via a flexible utility function $U_{metric}$, the framework can also be made compatible to fairness or diversity objectives. Together, it lays foundation for more reliable and trustworthy sequential recommender systems.

## 8 ETHICS STATEMENT

All datasets used in the work are publicly available, and the code repository is anonymized; no personal identifying information is involved. The study was conducted in accordance with guidelines for responsible research and reproducible science.

## 9 REPRODUCIBILITY STATEMENT

To facilitate reproducibility, we provide the following resources. 1) Source code and datasets: An anonymized implementation of our proposed framework, supporting codes and datasets are included in the anonymous repository. https://anonymous.4open.science/r/SURE_-02D2 2) Proofs: Formal statements and complete proofs underpinning our framework are provided in Section A.4 in the Appendix. 3) Hyperparameters and Implementation Details: The detailed implementation details and configurations are present in Section A.5 in the Appendix.

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

# A APPENDIX

## A.1 SUMMARY OF NOTATIONS

To facilitate clarity, we provide a comprehensive summary of the key mathematical notations and variables used throughout the SURE framework in Table 4.

## A.2 ASSUMPTIONS

We state two mild assumptions that we use in Theorems 5.1 and 5.2.

Table 4: Summary of Notations

| Symbol | Description |
|--------|-------------|
| $\mathcal{U}, \mathcal{I}$ | Sets of users and items |
| $u, i$ | Individual user and item |
| $\mathcal{H}_u$ | Interaction history for user $u$ |
| $i_{rel}^{t+1}$ | The true relevant next item at time $t+1$ |
| $L$ | Total number of base models (experts) in the ensemble |
| $\mathcal{M}^\ell$ | The $\ell$-th base recommender model ($\ell \in \{1, \dots, L\}$) |
| $\mathbf{w}^t$ | Ensemble weight vector at time $t$ ($\mathbf{w}^t \in \Delta^L$) |
| $s_\ell^t, S_\ell^t$ | Instantaneous and cumulative prediction set size for model $\ell$ |
| $\lambda_\ell^t$ | Calibration threshold for model $\ell$ at time $t$ |
| $\mathcal{C}_\ell^{t+1}$ | Prediction set generated by model $\ell$ using threshold $\lambda_\ell^t$ |
| $\mathcal{C}^{\mathrm{agg}}$ | Final aggregated ensemble prediction set |
| $\alpha$ | User-defined target error rate (risk level) |
| $R(\mathcal{C})$ | True risk of the prediction set |
| $\bar{R}_\ell^t$ | Average empirical risk over the current stable window $\mathcal{W}_\ell^t$ |
| $\mathcal{L}_u(\cdot)$ | Utility-based Risk Loss (e.g., $1 - $ Recall), used for calibration |
| $L_t(\cdot)$ | Predictive Loss (e.g., Cross-Entropy), used for shift detection |
| $d_\ell^{\mathrm{pref}}$ | Loss-Based Preference Shift Metric |
| $d_\ell^{\mathrm{ldd}}$ | Loss Discrepancy Distance (LDD) |
| $c_\ell^t$ | Start time of the current stable segment for model $\ell$ |
| $\mathcal{W}_\ell^t$ | Current stable window $[c_\ell^t, t]$ |
| $\eta$ | Hedge learning rate for updating ensemble weights |
| $\beta$ | Shift sensitivity parameter for change-point detection |
| $\gamma$ | Segment-length bias parameter for change-point detection |
| $\rho$ | Step size for the adaptive threshold update |

**Assumption A.1.** *For every base model $\mathcal{M}^\ell$ and any segment $\mathcal{W}_t^\ell$ produced by the change–point detector, there exists a threshold $\lambda_\ell^{\min} \in \Lambda$ such that*

$$R\big(\mathcal{C}_{\lambda_\ell^{\min}}^\ell\big) \leq \alpha.$$

*Equivalently, the mapping $\lambda \mapsto R(\mathcal{C}_\lambda^\ell)$ is continuous and attains all values in $[0, 1]$ on the closed set $\Lambda$.*

This assumption ensures that for every timestamp in each segment, it is possible to achieve risk control at level $\alpha$ by appropriately tuning $\lambda_\ell^t$. It guarantees the effectiveness of the update rule in Eq. 15.

**Assumption A.2.** *For each base model $\mathcal{M}^\ell$, let $\mathcal{W}_t^\ell = [c_t^\ell, t]$ denote the segment window at time $t$ returned by the Bayesian change-point detector. We assume the per-user utility losses $\big\{ \mathcal{L}_u\big(\mathcal{C}_{\lambda_\ell}^\ell\big) \big\}_{\tau \in \mathcal{W}_t^\ell, \, u \in \mathcal{U}_\tau}$ are drawn from a common bounded distribution within each segment. In other words, the loss values within $\mathcal{W}_t^\ell$ are exchangeable and lie in $[0, 1]$.*

This assumption allows average window risk $\bar{R}_t^\ell$ to serve as a faithful estimate of true segment risk.

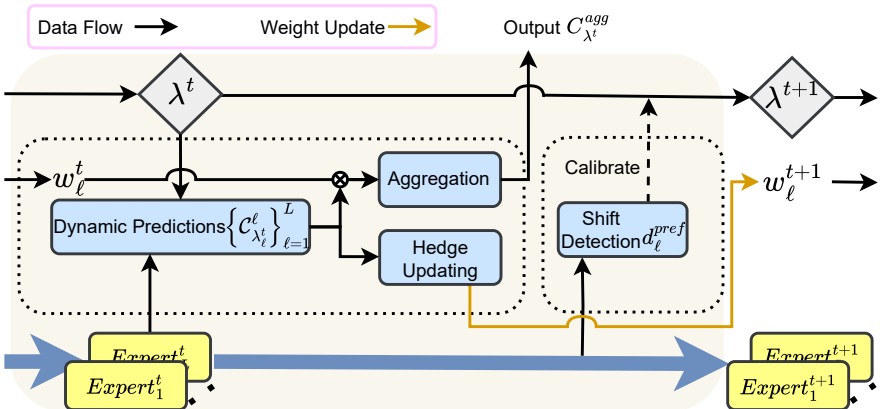

Figure 1: **The SURE WorkFlow (one-step update).** At each timestamp $t$, base experts produce dynamic prediction sets $\{\mathcal{C}^{\ell}_{\lambda^t_{\ell}}\}^{L}_{\ell=1}$ using thresholds $\lambda^t_{\ell}$. These sets are aggregated with Hedge weights $\mathbf{w}^t$ to form the ensemble recommendation $\mathcal{C}^{\text{agg}}_{\lambda^t}$. Shift detection computes preference change scores $d^{\text{pref}}_{\ell}$ and triggers recalibration, updating thresholds $\lambda^{t+1}_{\ell} \leftarrow \lambda^t_{\ell} - \rho(\bar{R}^{\ell}_t - \alpha)$, while Hedge updating adjusts the weights $\mathbf{w}^{t+1}$ based on set efficiency. The outputs $(\mathcal{C}^{\text{agg}}_{\lambda^t}, \lambda^{t+1}, \mathbf{w}^{t+1})$ are then passed to the next step, ensuring validity, compactness, and robustness over time.

## A.3 ALGORITHM

This section we provide the pseudocode for the *Dynamically Adaptive Uncertainty-aware Optimization (DAUO)* algorithm. The algorithm begins with an initial calibration phase where each base model $\mathcal{M}^{\ell}$ is assigned a starting threshold $\lambda^0_{\ell}$ and an equal ensemble weight. This initialization returns updated $\lambda^0_{\ell}$ which creates prediction sets (Eq. (2)) that ensure the empirical risk $\hat{R}(\mathcal{C}^{\ell}_{\lambda^0_{\ell}})$, estimated via Eq. (4), falls below the user-defined margin $(\alpha - \epsilon)$. Then at each timestamp $t$, DAUO adapts both the calibration threshold and ensemble weights in an online manner. For each base model $\mathcal{M}^{\ell}$, the algorithm first evaluates user preference shift by computing the divergence $d^{\text{pref}}_{\ell}(c^{\ell}_t, t)$ as per Eq. (12), followed by the Bayesian posterior over candidate segment boundaries using Eq. (13). Here $c^{\ell}_t$ represents all the timestamps after the last changepoint detected (where the framework predicted the preference shift). The segment start is then updated by selecting the most likely boundary $c^{\ell}_{t+1}$, and the average risk $\bar{R}^{\ell}_t$ over the new segment window $[c^{\ell}_{t+1}, t]$ is computed via Eq. (14). The threshold $\lambda^t_{\ell}$ is then updated according to Eq. (15), which adjusts the confidence level based on segmental risk deviation from $\alpha$. After all models have updated their thresholds, the ensemble weight vector $\mathbf{w}^t$ is revised via Eq. (9), giving higher weight to models producing more compact prediction sets. At prediction time $T+1$, the calibrated thresholds $\hat{\lambda}^T_{\ell}$ and final weights $\mathbf{w}^T$ are used to construct individual model prediction sets via Eq. (2). These are then merged through the weighted majority aggregation rule $\mathcal{A}(\cdot, \mathbf{w}^T)$ in Eq. (8) to produce the final ensemble prediction $\mathcal{C}^{\text{agg}}_{\hat{\lambda}^T}(\mathcal{H}_u)$. The detailed steps are presented in Algorithm 1.

## A.4 PROOFS

### A.4.1 THEOREM 1

**Lemma A.4.1**. *Let for each base model $\mathcal{M}^{\ell}$, the prediction set at round $t$ is $\mathcal{C}^{\ell}_{\lambda^t} \subset \mathcal{I}$ with $s^t_{\ell} := \left|\mathcal{C}^{\ell}_{\lambda^t}\right|$. Also, let $\mathbf{w}^t = (w^t_1, \ldots, w^t_L) \in \Delta^L$ and $k(t) \sim \text{Uniform}[0, 1]$. Given, the aggregated prediction set is defined by Eq. (7), we have:*

$$\mathbb{E}_{k(t)} \left[ \left| \mathcal{C}^{\text{agg}}_{\lambda^t} \right| \right] \le h_t$$

---

$\dagger\, Here, \Delta\lambda^{\dagger}$ is equivalent to $\lambda^t_{\ell}/|\Lambda|$, where $\Lambda$ is the set of candidate thresholds.

---

**Algorithm 1** Dynamically Adaptive Uncertainty-aware Optimization (DAUO)

---

1: **Initialization:**
2: Initialize thresholds $\lambda_\ell^0$ and ensemble weights $\mathbf{w}^0 = \frac{1}{L}\mathbf{1}$ for all base models $\ell = 1, \ldots, L$
3: Set user-defined parameters: target risk $\alpha$, confidence $\delta$, error tolerance $\epsilon$
4: Define utility-based loss as in Eq. (5)
5: Initialize per-model segment starts $c_\ell^0 = 1$

6: **for** each base model $\ell = 1, \ldots, L$ **do**
7:     Compute prediction set $\mathcal{C}_\ell^0$ using Eq. (2) with threshold $\lambda_\ell^0$
8:     Compute empirical risk $\hat{R}(\mathcal{C}_\ell^0)$ using Eq. (4)
9:     **if** $\hat{R}(\mathcal{C}_\ell^0) \leq \alpha - \epsilon$ **then**
10:        **continue**
11:    **else**
12:        Update threshold: $\lambda_\ell^0 \leftarrow \lambda_\ell^0 - \Delta\lambda^\dagger$
13:    **end if**
14: **end for**

15: **Calibration:**
16: **for** each timestamp $t = 1, \ldots, T$ **do**
17:    **for** each base model $\ell = 1, \ldots, L$ **do**
18:        Compute preference shift $d_\ell^{\text{pref}}(c_\ell^{t-1}, t)$ using Eq. (12)
19:        Compute posterior $p_\ell(c_\ell^t = c_\ell^{t-1} + k \mid t)$ using Eq. (13)
20:        Update segment start:

$$c_\ell^t \leftarrow \arg\max_k p_\ell(c_\ell^t = c_\ell^{t-1} + k \mid t)$$

21:        Compute window risk $\bar{R}_\ell^t$ on $[c_\ell^t, t]$ using Eq. (14)
22:        Update threshold using Eq. (15):

$$\lambda_\ell^{t+1} = \lambda_\ell^t - \rho\left(\bar{R}_\ell^t - \alpha\right)$$

23:    **end for**
24:    Update ensemble weights $\mathbf{w}^{t+1}$ using Eq. (9)
25: **end for**
26: Store final thresholds: $\hat{\lambda}_\ell^T \leftarrow \lambda_\ell^T$ and weights $\mathbf{w}^T$

27: **Output at timestamp** $T + 1$**:**
28: **for** each user $u \in \mathcal{U}$ **do**
29:    **for** each base model $\ell = 1, \ldots, L$ **do**
30:        Compute prediction set: $\mathcal{C}_\ell^T$ using Eq. (2) with threshold $\hat{\lambda}_\ell^T$
31:    **end for**
32:    Aggregate ensemble prediction sets using Eq. (8):

$$\mathcal{C}_{\hat{\boldsymbol{\lambda}}^T}^{\text{agg}}(\mathcal{H}_u) = \mathcal{A}\left(\{\mathcal{C}_\ell^T\}_{\ell=1}^L, \mathbf{w}^T\right)$$

33: **end for**

---

*where* $h_t := \sum_{\ell=1}^L w_\ell^t s_\ell^t$.

*Proof.* For any item $i \in \mathcal{I}$, the aggregated support on the item can be defined as:

$$\bar{w}_t(i) = \sum_{\ell=1}^L w_\ell^t \cdot \mathbf{1}\left[i \in \mathcal{C}_{\lambda^t}^\ell\right] \in [0, 1] \tag{i}$$

where $\bar{w}_t(i)$ is the total weight of models that include item $i$ in the prediction set.

By the definition of the aggregation rule, item $i$ is included in the aggregated prediction set iff:

$$\bar{w}_t(i) > \frac{1 + k(t)}{2} \qquad \text{(equivalently)} \tag{ii}$$

or equivalently

$$k(t) < 2\bar{w}_t(i) - 1 \tag{iii}$$

So the probability that item $i$ is in ensemble set is,

$$\Pr_{k(t)}\left[i \in \mathcal{C}^{\text{agg}}_{\lambda^t}\right] = \Pr_{k(t)}\left[k(t) < 2\bar{w}_t(i) - 1\right] \tag{iv}$$

Since $k(t) \sim \text{Uniform}[0,1]$, we know that

$$\Pr\left[k(t) < u\right] = \begin{cases} 0 & \text{if } u \leq 0 \\ u & \text{if } 0 < u < 1 \\ 1 & \text{if } u \geq 1 \end{cases} \qquad \text{for any real } u \tag{v}$$

Applying $u := 2\bar{w}_t(i) - 1$, then:

$$\Pr_{k(t)}\left[i \in \mathcal{C}^{\text{agg}}_{\lambda^t}\right] = (2\bar{w}_t(i) - 1)_+ \tag{vi}$$

with $(x)_+ := \max\{x, 0\}$.

Now we know for all $x \in [0, 1]$,

$$(2x - 1)_+ \leq x \quad \text{for x} \in (0, 1] \tag{vii}$$

i.e., if $x \leq \frac{1}{2}$, then $(2x - 1) \leq 0 \Rightarrow (2x - 1)_+ = 0$

if $x \geq \frac{1}{2}$, then $(2x - 1)_+ = 2x - 1$

and $(2x - 1) \leq x \Leftrightarrow x \leq 1$   [true]

Hence we can write

$$\Pr_{k(t)}\left[i \in \mathcal{C}^{\text{agg}}_{\lambda^t}\right] = (2\bar{w}_t(i) - 1)_+ \leq \bar{w}_t(i) \tag{viii}$$

Now computing the expected total size of ensemble set, we have:

$$\mathbf{E}_{k(t)}\left[|\mathcal{C}^{\text{agg}}_{\lambda^t}|\right] = \sum_{i \in \mathcal{I}} \Pr_{k(t)}\left[i \in \mathcal{C}^{\text{agg}}_{\lambda^t}\right] \leq \sum_{i \in \mathcal{I}} \bar{w}_t(i) \tag{ix}$$

From Eq. (i), and expanding $\bar{w}_t(i)$, we get:

$$\sum_{i \in \mathcal{I}} \bar{w}_t(i) = \sum_{i \in \mathcal{I}} \sum_{\ell=1}^{L} w^t_\ell \cdot \mathbf{1}\left[i \in \mathcal{C}^\ell_{\lambda^t}\right] \tag{x}$$

Switching the summation order, we get:

$$= \sum_{\ell=1}^{L} w^t_\ell \sum_{i \in \mathcal{I}} \mathbf{1}\left[i \in \mathcal{C}^\ell_{\lambda^t}\right] = \sum_{\ell=1}^{L} w^t_\ell \cdot \left|\mathcal{C}^\ell_{\lambda^t}\right| = \sum_{\ell=1}^{L} w^t_\ell \cdot s^t_\ell = h_t \tag{xi}$$

Putting this all together, we get:

$$\boxed{\mathbf{E}_{k(t)}\left[|\mathcal{C}^{\text{agg}}_{\lambda^t}|\right] \leq h_t := \sum_{\ell=1}^{L} w^t_\ell s^t_\ell} \tag{xii}$$

Hence Proved.

$\square$

**Remark** Lemma A.4.1 shows that the expected size of the aggregated prediction set is no greater than the surrogate size $h_t$, which is a weighted average of base model set sizes. This means the aggregation step does not inflate the prediction set and adapts to the ensemble's diversity at time $t$.

**PROOF OF THEOREM 5.1**

*Proof.* From Lemma 1, we already have:

$$\mathbf{E}_{k(t)}\left[\left|\mathcal{C}^{\mathrm{agg}}_{\lambda^t}\right|\right] \le h_t := \sum_{\ell=1}^{L} w_\ell^t s_\ell^t$$

Let

$$\hat{s}_\ell^t := \frac{s_\ell^t}{|\mathcal{I}|}, \qquad \hat{h}_t := \sum_{\ell=1}^{L} w_\ell^t \hat{s}_\ell^t = \frac{h_t}{|\mathcal{I}|} \quad \Rightarrow \quad h_t = |\mathcal{I}| \cdot \hat{h}_t$$

Now our goal is to bound $\hat{h}_t$ in terms of the best expert's size $\hat{s}_{\ell*}^t$. However, given the weights are spread across all $L$ models, we cannot directly bound $\hat{h}_t$. Taking inspiration from Cesa-Bianchi & Lugosi (2006); De Rooij et al. (2014), we analyze it via an auxiliary quantity called mix loss. Specifically, we decompose the Hedge average into two components: 1) mix loss that behaves like a soft minimum, and 2) a mixability gap that measures how far the weighted average is from the mix loss.

We first define mix loss as:

$$m_t := -\frac{1}{\eta} \log \sum_{\ell=1}^{L} w_\ell^t \cdot e^{-\eta \hat{s}_\ell^t} \tag{i}$$

and mixability gap as:

$$\delta_t := \hat{h}_t - m_t \quad \Rightarrow \quad \hat{h}_t = m_t + \delta_t \tag{ii}$$

To bound the mixability gap $\delta_t$, we use Bernstein's Cumulant Generating Function inequality:

Using (i) in (ii), we get:

$$\delta_t = \hat{h}_t + \frac{1}{\eta} \log \sum_{\ell=1}^{L} w_\ell^t \cdot e^{-\eta \hat{s}_\ell^t} \tag{iii}$$

Refactoring:

$$e^{-\eta \hat{s}_\ell^t} = e^{-\eta(\hat{s}_\ell^t - \hat{h}_t)} \cdot e^{-\eta \hat{h}_t} \tag{iv}$$

So,

$$\sum_\ell w_\ell^t \cdot e^{-\eta \hat{s}_\ell^t} = \sum_\ell w_\ell^t \cdot \left(e^{-\eta(\hat{s}_\ell^t - \hat{h}_t)} \cdot e^{-\eta \hat{h}_t}\right) = e^{-\eta \hat{h}_t} \cdot \sum_\ell w_\ell^t \cdot e^{-\eta(\hat{s}_\ell^t - \hat{h}_t)} \tag{v}$$

Plugging into log we get:

$$\log \sum_\ell w_\ell^t \cdot e^{-\eta \hat{s}_\ell^t} = -\eta \hat{h}_t + \log \sum_\ell w_\ell^t \cdot e^{-\eta(\hat{s}_\ell^t - \hat{h}_t)} \tag{vi}$$

Putting Eq. (vi) in Eq. (iii), we get:

$$\delta_t = \hat{h}_t + \frac{1}{\eta}\left(-\eta \hat{h}_t + \log \sum_\ell w_\ell^t \cdot e^{-\eta(\hat{s}_\ell^t - \hat{h}_t)}\right)$$

$$= \hat{h}_t - \hat{h}_t + \frac{1}{\eta} \log \sum_\ell w_\ell^t \cdot e^{-\eta(\hat{s}_\ell^t - \hat{h}_t)}$$

$$\delta_t = \frac{1}{\eta} \log \mathbf{E}_{\ell \sim w^t}\left[e^{-\eta(\hat{s}_\ell^t - \hat{h}_t)}\right] \tag{vii}$$

Now to bound $\delta_t$, we use the Bernstein Cumulant Generating Function (CGF) as introduced in Cesa-Bianchi & Lugosi (2006). We interpret $\hat{s}_\ell^t \in [0,1]$ as a bounded random variable under distribution $\ell \sim \mathbf{w}^t$, and apply the cumulant inequality.

Specifically, since $X := \hat{s}_\ell^t$,

$$\mathbf{E}[X] = \hat{h}_t, \quad \mathrm{Var}(X) = v_t$$

Now, defining moment generating function as:

$$\phi(\eta) := \log \mathbf{E}_{\ell \sim w^t} \left[ e^{-\eta(X - \mathbf{E}[X])} \right] \tag{viii}$$

Applying the result from Cesa-Bianchi & Lugosi (2006) for $\eta \in (0, 1]$ and any $X \in [0, 1]$, we get:

$$\log \mathbf{E} \left[ e^{-\eta(X - \mathbf{E}[X])} \right] \leq \frac{e^\eta - \eta - 1}{\eta} \cdot \mathrm{Var}(X) \tag{ix}$$

Applying this $\phi(\eta)$ into Eq. (vii) to Eq. (viii), we have:

$$\delta_t = \frac{1}{\eta} \cdot \phi(\eta) \tag{x}$$

And then applying the CGF bound we have:

$$\delta_t \leq \frac{1}{\eta} \cdot \left( \frac{e^\eta - \eta - 1}{\eta} \right) v_t = \frac{e^\eta - \eta - 1}{\eta^2} \cdot v_t \tag{xi}$$

Using Taylor series, we know:

$$e^\eta = 1 + \eta + \frac{\eta^2}{2} + \frac{\eta^3}{3!} + \dots$$

Simplifying, we get:

$$\delta_t \leq \left( \frac{\eta}{2} + \frac{\eta^2}{6} \right) v_t \tag{xii}$$

Next we need to bound the mix loss $m_t$

Given $\ell^* := \arg\min_{\ell \in [L]} \hat{s}_\ell^t$, we apply the classic log-sum-exp inequality:

For any real values $x_1, \dots, x_L$,

$$\log \sum_{\ell=1}^{L} e^{-x_\ell} \leq -\min_\ell x_\ell + \log L \tag{xiii}$$

Applying this to our case, with $x_\ell := \eta \hat{s}_\ell^t$, we get:

$$\log \sum_{\ell=1}^{L} e^{-\eta \hat{s}_\ell^t} \leq -\eta \hat{s}_{\ell^*}^t + \log L \tag{xiv}$$

As weight $\mathbf{w}^t \in \Delta^L$, the weighted sum is less than or equal to the uniform sum, i.e.,

$$\sum_{\ell=1}^{L} w_\ell^t \cdot e^{-\eta \hat{s}_\ell^t} \leq \sum_{\ell=1}^{L} e^{-\eta \hat{s}_\ell^t}$$

Hence, we get:

$$\log \sum_{\ell=1}^{L} w_\ell^t \cdot e^{-\eta \hat{s}_\ell^t} \leq \log \sum_{\ell=1}^{L} e^{-\eta \hat{s}_\ell^t} \leq -\eta \hat{s}_{\ell^*}^t + \log L \tag{xv}$$

Multiplying (xv) by $-\frac{1}{\eta}$, and applying a looser (but convenient) upper bound, we get:

$$m_t := -\frac{1}{\eta} \log \sum_{\ell=1}^{L} w_\ell^t \cdot e^{-\eta \hat{s}_\ell^t} \leq \hat{s}_{\ell^*}^t + \frac{\log L}{\eta} \tag{xvi}$$

From (xii) and (xvi), we get bounds for the mixability gap $\delta_t$ and mix loss $m_t$. Putting the results into Eq. (ii), we get:

$$\hat{h}_t \leq \hat{s}_{\ell^*}^t + \frac{\ln L}{\eta} + \left(\frac{\eta}{2} + \frac{\eta^2}{6}\right) v_t \tag{xvii}$$

Now we find the best $\eta$ that minimizes RHS in Eq. (xvii).

Let

$$f(\eta) := \frac{\ln L}{\eta} + \left(\frac{\eta}{2} + \frac{\eta^2}{6}\right) v_t$$

To minimize, we take derivative:

$$f'(\eta) = -\frac{\ln L}{\eta^2} + \left(\frac{1}{2} + \frac{\eta}{3}\right) v_t \tag{xviii}$$

Setting $f'(\eta) = 0$ and multiplying both sides by $\eta^2$, we get:

$$\frac{1}{2}\eta^2 + \frac{1}{3}\eta^3 = \frac{\ln L}{v_t} \tag{xix}$$

Since it is in cubic form, we approximate, getting:

$$\eta^* = \sqrt{\frac{2 \ln L}{v_t}}^{\ddagger}$$

Putting the $\eta^*$ in Eq. (xvii), and approximating, we get:

$$\hat{h}_t \leq \hat{s}_{\ell^*}^t + \sqrt{2 \ln L \cdot v_t} + \frac{2}{3} \ln L \tag{xx}$$

Now we know:

$$h_t = |\mathcal{I}| \cdot \hat{h}_t \quad \text{and} \quad s_{\ell^*}^t = |\mathcal{I}| \cdot \hat{s}_{\ell^*}^t$$

and

$$\mathbf{E}_{k(t)} \left[ \left| \mathcal{C}_{\lambda^t}^{\text{agg}} \right| \right] \leq h_t$$

Hence, we get:

$$\boxed{\mathbf{E}_{k(t)} \left[ \left| \mathcal{C}_{\lambda^t}^{\text{agg}} \right| \right] \leq s_{\ell^*}^t + \sqrt{2 \ln L \cdot v_t} + \frac{2}{3} \ln L}$$

Hence Proved.

$\square$

---

$\ddagger$ If $v_t = 0$, then all $\hat{s}_\ell^t$ are equal, so $\hat{h}_t = \hat{s}_{\ell^*}^t$, and the bound holds exactly. In this case, the variance penalty vanishes, and $\eta$ can be set arbitrarily (e.g., $\eta = 1$).

### A.4.2 THEOREM 2

**Lemma A.4.2**. *Let $\mathcal{M}^\ell$ be a base predictor and $\mathcal{U}_t^{\mathrm{cal}}$ be a batch of users at time $t$, with $n = |\mathcal{U}_t^{\mathrm{cal}}|$.*

*Assume for each user $u \in \mathcal{U}_t^{\mathrm{cal}}$, we observe the score $Z_{t,u}^\ell := \mathcal{M}^\ell(i_{\mathrm{rel}}^{t+1}(u) \mid \mathcal{H}_u^t)$, where the scores are sampled from a continuous distribution. Let $\lambda_t^\ell$ be the empirical $(1 - \alpha/2)$-quantile of the scores $\{Z_{t,u}^\ell\}_u$. Given the prediction set $\mathcal{C}_{\lambda_t^\ell}^\ell$ and the utility-based loss $\mathcal{L}_u(\mathcal{C}_{\lambda_t^\ell}^\ell)$ as defined in Eq. (4), then with probability at least $1 - \frac{1}{2n}$, over the calibration batch, the expected loss satisfies:*

$$\mathbb{E}_u\left[\mathcal{L}_u\left(\mathcal{C}_{\lambda_t^\ell}^\ell\right)\right] \leq \frac{\alpha}{2} + \sqrt{\frac{\log(4|\mathcal{U}_t^{\mathrm{cal}}|)}{2|\mathcal{U}_t^{\mathrm{cal}}|}}.$$

*Proof.* Given $n = |\mathcal{U}_t^{\mathrm{cal}}|$, let $Z_{t,u}^\ell \sim F$ for $u \in \mathcal{U}_t^{\mathrm{cal}}$, where $F$ is a continuous cumulative distribution function. We define the empirical CDF as:

$$\widehat{F}(z) := \frac{1}{n} \sum_{u \in \mathcal{U}_t^{\mathrm{cal}}} \mathbf{1}\left\{Z_{t,u}^\ell \leq z\right\}, \tag{i}$$

where $n = |\mathcal{U}_t^{\mathrm{cal}}|$.

Let $\lambda_t^\ell$ denote the empirical $(1 - \alpha/2)$-quantile of the scores $\{Z_{t,u}^\ell\}$, so by construction:

$$\widehat{F}(\lambda_t^\ell) \geq 1 - \frac{\alpha}{2}. \tag{ii}$$

To control the deviation between $\widehat{F}(\cdot)$ and the true CDF $F(\cdot)$, we apply the Dvoretzky–Kiefer–Wolfowitz (DKW) inequality:

For any $\varepsilon > 0$, we have:

$$\Pr\left(\sup_{z \in \mathbb{R}} \left|\widehat{F}(z) - F(z)\right| > \varepsilon\right) \leq 2\exp(-2\varepsilon^2 n). \tag{iii}$$

To ensure failure probability at most $\frac{1}{2n}$, we set:

$$2\exp(-2\varepsilon^2 n) = \frac{1}{2n}.$$

Solving this gives:

$$\varepsilon = \sqrt{\frac{\log(4n)}{2n}}. \tag{iv}$$

Using Eq. (iii), this gives a uniform deviation bound that holds with probability at least $1 - \frac{1}{2n}$.

From the DKW result, we have the uniform deviation bound:

$$\left|\widehat{F}(z) - F(z)\right| \leq \sqrt{\frac{\log(4n)}{2n}} \quad \text{for all } z \in \mathbb{R}. \tag{v}$$

Now at $z = \lambda_t^\ell$, we get:

$$F(\lambda_t^\ell) \geq \widehat{F}(\lambda_t^\ell) - \sqrt{\frac{\log(4n)}{2n}} \geq 1 - \frac{\alpha}{2} - \sqrt{\frac{\log(4n)}{2n}}. \tag{vi}$$

Hence, for a user sampled independently from the distribution, the score $Z_{t,u}^\ell \sim F$, and the probability that the true item is excluded from the prediction set is:

$$\Pr(Z_{t,u}^\ell > \lambda_t^\ell) = 1 - F(\lambda_t^\ell) \leq \frac{\alpha}{2} + \sqrt{\frac{\log(4n)}{2n}}.$$

Given the utility definition from Eq. (4) in main pasper, $\mathcal{L}_u(\mathcal{C}_{\lambda_t^\ell}) = 1$ when the true item is excluded. Thus, the expected utility loss for the user is:

$$\mathbb{E}_u\left[\mathcal{L}_u(\mathcal{C}_{\lambda_t^\ell})\right] \leq \left(1 - \frac{1}{2n}\right)\left(\frac{\alpha}{2} + \sqrt{\frac{\log(4n)}{2n}}\right) + \frac{1}{2n}.$$

For $n > 1$, i.e., at least 1 user in the calibration batch, $\frac{1}{2n} \leq \sqrt{\frac{\log(4n)}{2n}}$. For simplicity, we absorb the additive constant in the existing slack and simplify. Hence we get:

$$\mathbb{E}\left[\mathcal{L}_u(\mathcal{C}_{\lambda_t^\ell})\right] \leq \frac{\alpha}{2} + \sqrt{\frac{\log(4n)}{2n}}.$$

$$\boxed{\mathbb{E}\left[\mathcal{L}_u(\mathcal{C}_{\lambda_t^\ell})\right] \leq \frac{\alpha}{2} + \sqrt{\frac{\log(4n)}{2n}} := \frac{\alpha}{2} + \sqrt{\frac{\log\left(4|\mathcal{U}_t^{\text{cal}}|\right)}{2|\mathcal{U}_t^{\text{cal}}|}}.}$$

Hence Proved. $\qquad\square$

***Remark*** Lemma A.4.2 ensures that the utility-based loss of the prediction set $\mathcal{C}_{\lambda_t^\ell}^\ell$, estimated from a finite calibration batch, concentrates around the error level $\alpha/2$. As the calibration batch size $n \to \infty$, the slack term $\sqrt{\frac{\log(4n)}{2n}} \to 0$, the upper bound of expected loss achieves $\alpha/2$.

**Lemma A.4.3**. *Given $\mathcal{M}^\ell$ as a base model, let the change-point detector define a stable segment of timesteps $\mathcal{W}_t^\ell = [c_t^\ell, t]$, for which no user preference shift is detected. Let $\mathcal{L}_\tau^{(\ell)}(\mathcal{C}_{\lambda_\tau^\ell})$ denote the utility loss incurred by model $\mathcal{M}^\ell$ at time $\tau \in \mathcal{W}_t^\ell$. Given the empirical segment risk $\bar{R}_t^\ell$ as defined in Eq. (14), and let $\mathcal{F}_\tau$ denote the filtration capturing all user histories, model predictions, and losses observed up to time $\tau$, then for any $\epsilon > 0$, we have:*

$$\Pr\left(\bar{R}_t^\ell - \mathbb{E}\left[\bar{R}_t^\ell \mid \mathcal{F}_{c_t^\ell - 1}\right] \geq \epsilon\right) \leq \exp\left(-2\epsilon^2 |\mathcal{W}_t^\ell|\right).$$

*Proof.* Let $X_\tau$ define a random variable that captures the surprise at time $\tau \in \mathcal{W}_t^\ell$, i.e.,

$$X_\tau := \mathcal{L}_\tau^{(\ell)} - \mathbb{E}\left[\mathcal{L}_\tau^{(\ell)} \mid \mathcal{F}_{\tau-1}\right], \tag{i}$$

where $\mathcal{L}_\tau^{(\ell)}(\mathcal{C}_{\lambda_\tau^\ell})$ is the observed loss, and the expectation is our best guess before time $\tau$.

We now define the cumulative sum over $X_\tau$ as:

$$S_k := \sum_{\tau=c_t^\ell}^{k} X_\tau, \quad \text{for } k \in [c_t^\ell, t]. \tag{ii}$$

Now, the sequence $\{S_k\}$ is a martingale with respect to the filtration $\mathcal{F}_k$. Specifically:

$$\mathbb{E}[S_k \mid \mathcal{F}_{k-1}] = S_{k-1}. \tag{iii}$$

This relation holds because:

$$S_k = S_{k-1} + X_k \quad \Rightarrow \quad \mathbb{E}[S_k \mid \mathcal{F}_{k-1}] = S_{k-1} + \mathbb{E}[X_k \mid \mathcal{F}_{k-1}].$$

Now,

$$\mathbb{E}[X_k \mid \mathcal{F}_{k-1}] = \mathbb{E}\left[\mathcal{L}_k^\ell - \mathbb{E}\left[\mathcal{L}_k^{(\ell)} \mid \mathcal{F}_{k-1}\right] \mid \mathcal{F}_{k-1}\right] \tag{iv}$$

By linearity and the idempotence of conditional expectation, we directly get:

$$\mathbb{E}[L_k^\ell \mid \mathcal{F}_{k-1}] - \mathbb{E}[L_k^\ell \mid \mathcal{F}_{k-1}] = 0. \tag{v}$$

Hence $X_k$ is a martingale difference, and $\{S_k\}$ is a martingale.

Also, since $\mathcal{L}_\tau^\ell(\mathcal{C}_{\lambda_\tau^\ell}) \in [0, 1]$, its conditional expectation also lies in $[0, 1]$, and therefore:

$$|X_k| \leq 1 \quad \text{i.e., the increments are bounded.}$$

Now, by Azuma–Hoeffding's inequality, for any martingale with bounded increments $|X_k| \leq 1$, the following holds: From Azuma–Hoeffding's inequality, we now have:

$$\Pr(S_t \geq \epsilon | \mathcal{W}_t^\ell|) \leq \exp\left(-2\epsilon^2 |\mathcal{W}_t^\ell|\right), \tag{vi}$$

where $\epsilon > 0$, and $|\mathcal{W}_t^\ell| = t - c_t^\ell + 1$.

Now we relate $S_t$ to the definition of empirical risk. Given the definition of average risk over a window, we have:

$$\mathbb{E}\left[\bar{R}_t^\ell \mid \mathcal{F}_{c_t^\ell-1}\right] = \mathbb{E}\left[\frac{1}{w} \sum_{\tau=c_t^\ell}^t \mathcal{L}_\tau^\ell(\mathcal{C}_{\lambda_\tau^\ell}) \,\middle|\, \mathcal{F}_{c_t^\ell-1}\right]$$

$$= \frac{1}{w} \sum_{\tau=c_t^\ell}^t \mathbb{E}\left[\mathcal{L}_\tau^\ell(\mathcal{C}_{\lambda_\tau^\ell}) \,\middle|\, \mathcal{F}_{c_t^\ell-1}\right], \tag{vii}$$

where $w = |t - c_t^\ell + 1| := |\mathcal{W}_t^\ell|$.

Using the tower property of conditional expectation, for any $\tau \geq c_t^\ell$, we have:

$$\mathbb{E}\left[\mathcal{L}_\tau^\ell \mid \mathcal{F}_{c_t^\ell-1}\right] = \mathbb{E}\left[\mathbb{E}\left[\mathcal{L}_\tau^\ell \mid \mathcal{F}_{\tau-1}\right] \,\middle|\, \mathcal{F}_{c_t^\ell-1}\right]. \tag{viii}$$

Now, given the expression for deviation from expected risk:

$$\bar{R}_t^\ell - \mathbb{E}[\bar{R}_t^\ell \mid \mathcal{F}_{c_t^\ell-1}],$$

expanding this gives:

$$\frac{1}{w} \sum_{\tau=c_t^\ell}^t \mathcal{L}_\tau^\ell - \frac{1}{w} \sum_{\tau=c_t^\ell}^t \mathbb{E}\left[\mathcal{L}_\tau^\ell \mid \mathcal{F}_{\tau-1}\right].$$

Continuing from the previous expression, we now write:

$$\bar{R}_t^\ell - \mathbb{E}\left[\bar{R}_t^\ell \mid \mathcal{F}_{c_t^\ell-1}\right] = \frac{1}{w} \sum_{\tau=c_t^\ell}^t \left(\mathcal{L}_\tau^\ell - \mathbb{E}[\mathcal{L}_\tau^\ell \mid \mathcal{F}_{\tau-1}]\right). \tag{ix}$$

Now applying the tower property again, and using the result from Eq. (iv), we observe:

$$\mathcal{L}_\tau^\ell - \mathbb{E}\left[\mathbb{E}[\mathcal{L}_\tau^\ell \mid \mathcal{F}_{\tau-1}] \mid \mathcal{F}_{c_t^\ell-1}\right] = \mathbb{E}\left[X_\tau \mid \mathcal{F}_{c_t^\ell-1}\right]. \tag{x}$$

Putting Eq. (x) into Eq. (ix), we obtain:

$$\mathbb{E}\left[\bar{R}_t^\ell - \mathbb{E}\left[\bar{R}_t^\ell \mid \mathcal{F}_{c_t^\ell-1}\right]\right] = \frac{1}{w} \sum_{\tau=c_t^\ell}^t \mathbb{E}\left[X_\tau \mid \mathcal{F}_{c_t^\ell-1}\right]. \tag{xi}$$

Since we are bounding this deviation in probability, we retain the raw form:

$$\bar{R}_t^\ell - \mathbb{E}\left[\bar{R}_t^\ell \mid \mathcal{F}_{c_t^\ell-1}\right] = \frac{1}{w} \sum_{\tau=c_t^\ell}^t X_\tau = \frac{S_t}{w}. \tag{xii}$$

Now we finally substitute the result from Eq. (xii) into the Azuma–Hoeffding inequality Eq. (vi):

$$\Pr\left(\bar{R}_t^\ell - \mathbb{E}\left[\bar{R}_t^\ell \mid \mathcal{F}_{c_t^\ell - 1}\right] \geq \epsilon\right) = \Pr\left(\frac{S_t}{w} \geq \epsilon\right) \tag{xiii}$$

$$= \Pr\left(S_t \geq \epsilon w\right) \leq \exp\left(-2\epsilon^2 w\right). \tag{xiv}$$

Hence, we finally obtain the main result:

$$\boxed{\Pr\left(\bar{R}_t^\ell - \mathbb{E}[\bar{R}_t^\ell \mid \mathcal{F}_{c_t^\ell - 1}] \geq \epsilon\right) \leq \exp\left(-2\epsilon^2 \cdot |\mathcal{W}_t^\ell|\right)}$$

Hence Proved. □

**Remark** Lemma A.4.3 justifies using the empirical average risk $\bar{R}_t^\ell$ as a reliable proxy for the true conditional expectation and supports the adaptive threshold update rule in Eq. (15) of the framework.

**Corollary A.4.1.** *Given the threshold update rule from Eq. (15) of the framework:* $\lambda_\ell^{t+1} = \lambda_\ell^t - \rho\left(\bar{R}_t^{(\ell)} - \alpha\right)$, *then for any* $\delta \in (0, 1)$, *with probability at least* $1 - \delta$, *the deviation of the update from the ideal update satisfies:*

$$\left|\lambda_\ell^{t+1*} - \lambda_\ell^{t+1}\right| := \rho\left|\mathbb{E}[\bar{R}_t^\ell \mid \mathcal{F}_{c_t^\ell - 1}] - \bar{R}_t^\ell\right| \leq \rho \cdot \sqrt{\frac{\log(1/\delta)}{2|\mathcal{W}_t^\ell|}}.$$

*Proof.* From Lemma 2, with probability at least $1 - \delta$, we have:

$$\bar{R}_t^\ell - \mathbb{E}\left[\bar{R}_t^\ell \mid \mathcal{F}_{c_t^\ell - 1}\right] = \frac{S_t}{w} \quad \Rightarrow \quad \Pr\left(\bar{R}_t^\ell - \mathbb{E}[\bar{R}_t^\ell] \geq \epsilon\right) \leq \exp\left(-2\epsilon^2 w\right).$$

We now want to choose $\epsilon$ such that:

$$\exp\left(-2\epsilon^2 w\right) = \delta \quad \Rightarrow \quad \epsilon^2 = \frac{\log(1/\delta)}{2w} \quad \Rightarrow \quad \epsilon = \sqrt{\frac{\log(1/\delta)}{2w}} \tag{i}$$

Using Eq. (i), we can conclude that with probability at least $1 - \delta$:

$$\left|\bar{R}_t^\ell - \mathbb{E}[\bar{R}_t^\ell \mid \mathcal{F}_{c_t^\ell - 1}]\right| \leq \sqrt{\frac{\log(1/\delta)}{2w}}. \tag{ii}$$

Now substituting Eq. (ii) into the threshold update in framework's Eq. (15), and comparing with the ideal update:

$$\lambda_\ell^{t+1*} := \lambda_\ell^t - \rho\left(\mathbb{E}\left[\bar{R}_t^\ell \mid \mathcal{F}_{c_t^\ell - 1}\right] - \alpha\right),$$

we conclude that:

$$\boxed{\left|\lambda_\ell^{t+1*} - \lambda_\ell^{t+1}\right| := \rho\left|\mathbb{E}[\bar{R}_t^\ell \mid \mathcal{F}_{c_t^\ell - 1}] - \bar{R}_t^\ell\right| \leq \rho \cdot \sqrt{\frac{\log(1/\delta)}{2|\mathcal{W}_t^\ell|}}.}$$

Hence Proved. □

**Remark** From Corollary A.4.1 we observe that the adaptive threshold update remains close to its ideal value, even when using empirical segment risk. As the stable window length $|\mathcal{W}_t^\ell|$ increases, the deviation vanishes at a $O(1/\sqrt{|\mathcal{W}_t^\ell|})$ rate. This ensures the DAUO algorithm adapts reliably to user preferences over time, with provable statistical stability.

**Lemma A.4.4.** *Let* $\mathcal{M}^1, \ldots, \mathcal{M}^L$ *be L base models. Assume that for each model* $\mathcal{M}^\ell$, *the calibrated prediction set* $\mathcal{C}_{\lambda_\ell}^\ell t$ *satisfies the per-model miss probability bound:* $\Pr\left(i_{\text{rel}}^{t+1}(u) \notin \mathcal{C}_{\lambda_\ell^t}^\ell(S_u^t) \mid \mathcal{F}_{t-1}\right) \leq$

$\beta$ *for all $\ell = 1, \ldots, L$, where $\beta := \frac{\alpha}{2} + \varepsilon$,   and   $\varepsilon := \sqrt{\frac{\log(4|\mathcal{U}|)}{2|\mathcal{U}|}}$. Let $\mathcal{C}_{\boldsymbol{\lambda}^t}^{\mathrm{agg}}$ denote the ensemble prediction set formed by randomized weighted majority voting, using aggregation weights $\mathbf{w}^t \in \Delta^L$, the probability simplex.*

*Then the miss probability of the ensemble satisfies:*

$$\Pr\left(i_{\mathrm{rel}}^{t+1}(u) \notin \mathcal{C}_{\boldsymbol{\lambda}^t}^{\mathrm{agg}} \mid \mathcal{F}_{t-1}\right) \leq \alpha + 2\varepsilon.$$

*Proof.* For any user $u$, we define the miss indicator for model $\mathcal{M}^\ell$ as:

$$M_\ell := \mathbf{1}\left\{i_{\mathrm{rel}}^{t+1}(u) \notin \mathcal{C}_{\lambda_\ell^t}^{(\ell)}(S_u^{(t)})\right\}. \tag{i}$$

The ensemble predictor will fail if the true item receives insufficient support, i.e, the total weight of models that include the item is less than $\frac{1}{2}$. Equivalently, the total weight of models that miss the item exceeds $\frac{1}{2}$.

We formally define the total miss weight:

$$\sum_{\ell=1}^{L} w_\ell^t \cdot M_\ell. \tag{ii}$$

Then the ensemble misses if the above is $\geq \frac{1}{2}$. We wish to bound the probability of ensemble failure:

$$\Pr\left(\sum_{\ell=1}^{L} w_\ell^t \cdot M_\ell \geq \frac{1}{2} \;\middle|\; \mathcal{F}_{t-1}\right).$$

Applying Markov's inequality:

$$\Pr(X \geq a) \leq \frac{\mathbb{E}[X]}{a},$$

we obtain:

$$\Pr\left(\sum_{\ell=1}^{L} w_\ell^t \cdot M_\ell \geq \frac{1}{2} \;\middle|\; \mathcal{F}_{t-1}\right) \leq 2 \cdot \mathbb{E}\left[\sum_{\ell=1}^{L} w_\ell^t \cdot M_\ell \;\middle|\; \mathcal{F}_{t-1}\right]. \tag{iii}$$

Now, by linearity of expectation, we have:

$$\mathbb{E}\left[\sum_{\ell=1}^{L} w_\ell^t M_\ell \;\middle|\; \mathcal{F}_{t-1}\right] = \sum_{\ell=1}^{L} w_\ell^t \cdot \mathbb{E}\left[M_\ell \mid \mathcal{F}_{t-1}\right] = \sum_{\ell=1}^{L} w_\ell^t \cdot \Pr(M_\ell = 1 \mid \mathcal{F}_{t-1}). \tag{iv}$$

By Lemma A.4.2, each model satisfies:

$$\Pr(M_\ell = 1 \mid \mathcal{F}_{t-1}) \leq \beta. \tag{v}$$

Therefore,

$$\sum_{\ell=1}^{L} w_\ell^t \cdot \Pr(M_\ell = 1 \mid \mathcal{F}_{t-1}) \leq \beta \cdot \sum_{\ell=1}^{L} w_\ell^t = \beta. \tag{vi}$$

Substituting result from Eq. (vi) to Eq. (iii) back, we get the final ensemble miss bound:

$$\boxed{\Pr\left(i_{\mathrm{rel}}^{t+1}(u) \notin \mathcal{C}_{\boldsymbol{\lambda}^t}^{\mathrm{agg}} \mid \mathcal{F}_{t-1}\right) \leq 2\beta = \alpha + 2\varepsilon.} \tag{vii}$$

Hence Proved. $\qquad\square$

***Remark*** Lemma A.4.4 shows that the ensemble miss probability remains bounded by $\alpha + 2\varepsilon$ and preserves statistical validity despite possible correlation among predictors. As the calibration batch size $|\mathcal{U}| \to \infty$, the deviation $\varepsilon \to 0$, and the ensemble risk converges to $\alpha$.

**PROOF OF THEOREM 2**

*Proof.* Let $m := |\mathcal{U}|$ and $\varepsilon := \sqrt{\frac{\log(4m)}{2m}}$. Let $\mathcal{S} \subseteq \{1, \ldots, T\}$ denote the stable timestamps, where no preference shift is detected, and let $\mathcal{D} := \{1, \ldots, T\} \setminus \mathcal{S}$ denote the detection delay rounds. Then, we can say:

$$|\mathcal{S}| = T - D_T, \quad |\mathcal{D}| = D_T.$$

From Lemmas A.4.2 and A.4.4 , the expected loss satisfies:

$$\mathbb{E}\left[\mathcal{L}_u\left(\mathcal{C}_{\boldsymbol{\lambda^t}}^{\mathrm{agg}}\right) \mid \mathcal{F}_{t-1}\right] \leq \alpha + 2\varepsilon. \tag{i}$$

For $t \in \mathcal{D}$, the DAUO algorithm may be out-of-calibration. We conservatively assume the worst-case loss of 1 at each such round. There are $D_T$ such rounds yielding:

$$\sum_{t \in \mathcal{D}} \mathbb{E}\left[\mathcal{L}_u^{(t)}\right] \leq D_T. \tag{ii}$$

Now we handle the additional slack from DKW failures. At each round $t \in [T]$ and for each model $\ell \in [L]$, we calibrate the threshold using DKW. So there are $T \times L$ calibration events.

Let $Z_{t,\ell} \in \{0, 1\}$ be the indicator that DKW calibration fails at round $t$ for model $\ell$.

Then the total number of failures is:

$$K := \sum_{t=1}^{T} \sum_{\ell=1}^{L} Z_{t,\ell}. \tag{iii}$$

By Lemma A.4.2, each calibration failure has probability at most: $p := \frac{1}{2m}$. From Lemma 1, each DKW calibration failure has probability at most $p = \frac{1}{2m}$, and there are $T \times L$ such events. Thus, the expected number of failures is:

$$\mu := \mathbb{E}[K] = \frac{TL}{2m}.$$

We want to control the tail deviation:

$$\Pr(K \geq \mu + y) \leq \delta.$$

Using the Bernstein bound, we have:

$$\Pr(K \geq \mu + y) \leq \exp\left(\frac{-y^2}{2(\mu + y/3)}\right). \tag{iv}$$

To satisfy this inequality with probability $\geq 1 - \delta$, we choose $y$ to dominate both the average and tail slack. Following standard practice, we set:

$$y := \max\left\{\mu, \, 2\log\left(\tfrac{1}{\delta}\right)\right\}.$$

This guarantees:

$$\frac{y^2}{2(\mu + y/3)} \geq \log\left(\tfrac{1}{\delta}\right).$$

In realistic recommender settings, $m \gg L$, therefore:

$$\mu = \frac{TL}{2m} \leq 2\log\left(\tfrac{1}{\delta}\right).$$

Thus we may safely choose:

$$y = 2\log\left(\tfrac{1}{\delta}\right).$$

With this value, we get the high-probability bound:

$$K \leq \mu + y \leq \frac{TL}{2m} + 2\log\left(\tfrac{1}{\delta}\right). \tag{v}$$

Divide inequality (v) by $T$, we obtain:

$$\frac{K}{T} \leq \frac{TL}{2mT} + \frac{2\log(1/\delta)}{T}.$$

Since $\frac{TL}{2m} \leq 2\log(1/\delta)$ (by assumption), we get:

$$\frac{K}{T} \leq \frac{2\log(1/\delta)}{T}. \tag{vi}$$

Now combine the bounds from (i), (ii), and (vi):

$$\frac{1}{T}\sum_{t=1}^{T}\mathbb{E}\left[\mathcal{L}_u\left(\mathcal{C}_{\boldsymbol{\lambda}^t}^{\mathrm{agg}}\right)\right] \leq \frac{T-D_T}{T}(\alpha+2\varepsilon) + \frac{D_T}{T}\cdot 1 + \frac{K}{T}.$$

Substitute $\frac{K}{T} \leq \frac{2\log(1/\delta)}{T}$ and simplifying we get:

$$\frac{1}{T}\sum_{t=1}^{T}\mathbb{E}\left[\mathcal{L}_u\left(\mathcal{C}_{\boldsymbol{\lambda}^t}^{\mathrm{agg}}\right)\right] \leq \alpha + 2\varepsilon + \frac{D_T+2\log(1/\delta)}{T}. \tag{vii}$$

At round $T+1$, the ensemble prediction set $\mathcal{C}_{\boldsymbol{\lambda}^T}^{\mathrm{agg}}$ is formed using the thresholds $\boldsymbol{\lambda}^T$ trained across rounds 1 to $T$.

Assuming no additional change-point occurs at round $T+1$, a standard assumption in horizon-end guarantees, the loss distribution is equivalent to a stable round. Thus, the same bound applies, yielding:

$$\boxed{\mathbb{E}_{u\sim\mathcal{U}}\left[\mathcal{L}_u(\mathcal{C}_{\boldsymbol{\lambda}^T}^{\mathrm{agg}})\right] \leq \alpha + 2\sqrt{\frac{\log(4|\mathcal{U}|)}{2|\mathcal{U}|}} + \frac{D_T+2\log(1/\delta)}{T}.}$$

Hence Proved. $\qquad\qquad\square$

## A.5 IMPLEMENTATION DETAILS

In this section, we elaborate on the implementation details of the experiments conducted. The experiments were conducted on NVIDIA A40 GPU. Firstly, all base recommender models, NCF[19], CASER[39], SASRec[25], and FMLP-Rec[47] are trained for 100 epochs with a batch size of 256, a learning rate of 0.001, the Adam optimizer, and Binary Cross Entropy Loss (BCELoss). These models are implemented following their respective public repositories. User preference-aware baselines include TiSASRec[27], CDR[41], and Oracle4Rec[42]. TiSASRec extends SASRec with time-aware attention and relation-based temporal encoding, trained for 200 epochs with a batch size of 128. CDR employs a variational framework with domain-level disentanglement, trained for 200 epochs with a batch size of 512 and a learning rate of 0.0001. Oracle4Rec trains for 100 epochs with a batch size of 256 using a Transformer-style architecture with GELU activations and dropout regularization. These models retain their original optimization logic and regularization strategies. We furthermore implement three conformal prediction baselines: Split Conformal[40], EnbPI[43], and Online Conformal Prediction[1]. All conformal variants reuse the predicted score files from the base models and calculate expected loss based on ranking-based loss functions (e.g., MRR, NDCG, Recall). For Split Conformal, we determine the fixed prediction threshold via the $(1-\alpha)$-quantile of the first calibration timestamp, with $\alpha = 0.1$. For EnbPI, we use an ensemble of 10 bootstrapped recommendation models, with predictions aggregated using the sample mean. Prediction set widths were updated after each instance using a sliding window of the most recent $T = 5$ residuals. The miscoverage level was set to $\alpha = 0.1$, and expected loss was computed based on the same utility metrics. For Online Conformal Prediction, we use a decaying step size update rule, with the threshold updated after each instance. We set $\alpha = 0.1$ and used the same loss definitions as in other conformal methods explained above. The initial threshold $\lambda^0$ was shared across all conformal variants and our framework to ensure consistent initialization. Our proposed framework is implemented on top of the base recommendation model outputs. We conduct a manual search over the contrasting

hyperparameters in our Bayesian change-point module: the shift sensitivity $\beta \in \{0.5, 0.7, 0.9, 1.1\}$ and the segment-length bias $\gamma \in \{0, 0.3, 0.5, 0.7, 1, 1.3, 1.5, 1.75, 2\}$. Based on manual validation of segment stability and calibration smoothness across datasets, we fixed $\beta = 0.7$ and $\gamma = 1.1$. The error tolerance value $\epsilon$ is chosen based on the dataset size and the confidence value $\delta$. The threshold update step size $\eta$ in Eq. (15) was set to 0.05 throughout. To ensure consistency and reproducibility, we reused the predicted score files generated by the trained base models for all conformal baselines and our framework.

### A.5.1 UTILITY FUNCTION DEFINITIONS

The user utility function $U_{metric}(i_{rel}^{t+1}, \mathcal{C}_{\lambda^t})$, used in the loss formulation in Eq. (5) in main paper quantifies how well the prediction set $\mathcal{C}_{\lambda^t} \subseteq \mathcal{I}$ captures the relevant item $i_{rel}^{t+1}$ under different evaluation metrics. We define the following instantiations of $U_{metric}$ based on standard recommendation metrics:

**Recall-based utility:**
$$U_{\text{recall}}(i_{rel}^{t+1}, \mathcal{C}_{\lambda^t}) = \mathbb{I}[i_{rel}^{t+1} \in \mathcal{C}_{\lambda^t}]. \tag{viii}$$
This utility equals 1 if the relevant item is present in the prediction set and 0 otherwise.

**MRR-based utility:**
$$U_{\text{mrr}}(i_{rel}^{t+1}, \mathcal{C}_{\lambda^t}) = \begin{cases} \frac{1}{r(i_{rel}^{t+1})}, & \text{if } i_{rel}^{t+1} \in \mathcal{C}_{\lambda^t}, \\ 0, & \text{otherwise}, \end{cases} \tag{ix}$$
where $r(i_{rel}^{t+1})$ denotes the rank position of the relevant item within $\mathcal{C}_{\lambda^t}$, assuming items are ordered by decreasing model score.

**NDCG-based utility:**
$$U_{\text{ndcg}}(i_{rel}^{t+1}, \mathcal{C}_{\lambda^t}) = \frac{1}{\log_2(r(i_{rel}^{t+1}) + 1)} \cdot \mathbb{I}[i_{rel}^{t+1} \in \mathcal{C}_{\lambda^t}], \tag{x}$$
which discounts the gain based on the rank of the relevant item in the prediction set.

These definitions are used across all calibration and evaluation steps to compute utility-based loss values and coverage metrics.

### A.6 DETAILED EXPERIMENTATION DETAILS

In the main paper, we introduced five different datasets to evaluate the effectiveness of our framework. Below, we provide further details on the datasets, data-preprocessing, the base models, the user-preference aware baselines, and the conformal baselines used for comparison.

### A.6.1 DATASETS

- **Book-Crossing**[48]: a book-review dataset with explicit ratings and browsing logs.
- **Last.fm**[6]: music-streaming listening histories dataset providing implicit feedback.
- **Taobao**[22]: a large-scale e-commerce dataset with clicks, carts, and purchases attributes.
- **MovieLens**[18]: an explicit and implicit feedback dataset in the movie-rating domain.
- **Gowalla**[10]: a location-based social-network checkins dataset for point-of-interest recommendation.

All datasets are time-ordered, filtered using a 50-core strategy, and processed according to the data preprocessing and splitting procedure described below.

### A.6.2 SAMPLING AND DATA SPLITTING

- **Negative sampling.** Following the common experimentation strategy in recommendation frameworks, we select 50 non-interacted items per user at every time-stamp through negative sampling for training, validation, and testing.

- **Data Splitting.** Inspired by the sliding-window evaluation, we partition each dataset into five contiguous time-ordered batches $B_1, \ldots, B_5$ to capture potential shifts in user preferences over time. Within a batch, the first $80\%$ of interactions are used to train the model. The next $20\%$ are used to calibrate the conformal threshold $\lambda_\ell^t$ and weight parameters $\mathbf{w}^t$, while for the final interaction, the previously learned threshold and weight parameters are frozen and the framework is evaluated. The final results presented represent the average over all batches.

- **Multiple trials:** To account for variability in sampling, we repeat the experiments over 20 independent trials. For each trial, random negative samples were drawn for training, validation, and testing. The results were averaged across all the trials.

### A.6.3 BASE RECOMMENDATION MODELS

We build our framework on top of four representative recommendation backbones, each capturing different modeling paradigms:

- **Neural Collaborative Filtering (NCF)**[19]: Involves combination of GMF (Generalized Matrix Factorization) with 8-dimensional embeddings and MLP using layers $[64, 32, 16]$ with ReLU and dropout; combined with a prediction layer over concatenated representations.

- **Caser**[39]: A convolutional sequence model using vertical and horizontal filters with varying receptive fields over a fixed-length user interaction sequence. Configured with embedding dimension $d = 50$, sequence length $L = 5$, number of horizontal and vertical filters $n_h = 16$, $n_v = 4$, followed by a fully connected layer and dropout ($p = 0.5$).

- **SASRec**[25]: A Transformer-style sequential recommender with 2 self-attention blocks, 1 attention head, hidden size of 50, max sequence length of 50, and dropout rate of 0.5. Layer normalization, residual connections, and position encoding are used to model sequential dependencies.

- **FMLP-Rec**[47]: A Filter-Enhanced MLP model replacing attention heads with learned convolutional filters. Configured with hidden size of 64, 2 filter-enhanced encoder layers, 2 attention heads, dropout $= 0.5$, and GELU activation. Position embeddings and layer normalization are applied on top of the input sequence.

### A.6.4 PREFERENCE-AWARE RECOMMENDATION MODELS

To capture evolving user preferences and temporal context, we additionally incorporate three specialized preference-aware baselines:

- **TiSASRec:**[27] A time-aware sequential recommender model that extends SASRec by incorporating absolute and relative time information into the attention mechanism. We use 2 attention blocks, 1 attention head, and a hidden dimension of 50, along with a time matrix span of 256 and dropout rate of 0.2.

- **CDR (Causal Debiasing Recommendation):**[41] A user-centric causal recommendation model that disentangles user preferences across multiple training environments by learning group-invariant representations. We configure the MLP encoder as $[100, 20]$, preference encoder as $[100, 200]$, with latent variables all set to dimension 2. Dropout is set to 0.5 and batch norm is enabled.

- **Oracle4Rec:**[42] A 5-layer Transformer-style encoder with hidden size 128, 2 attention heads, GELU activation, and dropout of 0.5. It learns forward-looking user preferences by leveraging future interactions as oracle guidance. It employs two parallel encoders with shared embeddings: a Past Information Encoder and a Future Information Encoder, each comprising a noise filtering module, a causal self-attention module, and an interaction prediction layer.

### A.6.5 CONFORMAL PREDICTION BASELINES

We implemented three conformal prediction baselines and adapted them for recommendation tasks using ranking-based losses based on recommendation metrics(Recall, MRR, and NDCG). For each method, we used calibrated scores and constructed dynamic prediction sets over time.

- **Split Conformal Prediction:** A simple offline baseline where a global threshold $\lambda$ is computed and fixed during calibration and inference. Prediction sets are constructed by thresholding sorted item scores per user. This method serves as a non-adaptive control with no online feedback or user preference modeling.

- **Ensemble Batch Prediction Interval (EnbPI):** A time series conformal approach adapted for sequential recommendation task, uses a chosen sliding window of size 5 and a shift size $s=1$ for full online behavior. An ensemble of 10 base models is used, and the prediction sets are constructed by aggregating top items across models using a mean-based ensemble score. The threshold $\lambda$ is updated after each interaction using decayed step size based on loss deviations.

- **Online Conformal:** A fully online adaptive approach that dynamically recalibrates the threshold $\lambda$ based on user-specific risk feedback. After each interaction, the conformal predictor computes the empirical loss based on the utility metric and updates $\lambda^t$ using a gradient-based rule with decay. Like EnbPI, prediction sets are constructed using sorted calibrated scores, but don't use model ensembling.

### A.7 Additional Experiments

#### A.7.1 Results compared with base models and Preference-Aware baselines (Cont.)

We extend the analysis provided in the main paper, where we evaluate the SURE framework using four recommendation base models and against three user-preference-aware baselines in terms of recommendation metrics (i.e., MRR, Recall, NDCG). We present the results of the experimentations conducted on Taobao, MovieLens and Gowalla Datasets in Tables 5 and 6. These tables support the key findings: the SURE framework consistently controls risk within the predefined threshold $\alpha = 0.05$ with high confidence across all the base models, and as a result, it consistently outperforms all baselines on different performance metrics (MRR, Recall, NDCG) across datasets. This further validates the dataset-agnostic nature of our framework.

#### A.7.2 Results compared to Conformal baselines (Cont.)

Next, we continue our analysis comparing our framework with different conformal baselines in terms of coverage and set size. We conduct the experiments on Last.fM (Table 7), Taobao (Table 8), MovieLens (Table 9) and Gowalla (Table 10) datasets respectively and compare the results on base recommender models. The results reaffirm the main paper observations that our framework can ensure the best coverage–efficiency trade-off on every base model across datasets, ensuring valid recommendation sets.

#### A.7.3 Parameter Analysis

We analyze the influence of error rate $\alpha$, confidence parameter $\delta$, change-point detector parameters $(\beta, \gamma)$, and the number of experts $L$ on the recommendation sets generated by the SURE framework.

We first evaluate the impact of error rate $\alpha$, varying in $[0.05, 0.07, 0.10, 0.12, 0.15]$, on performance and the average prediction set sizes under fixed confidence thresholds $\delta = 0.05$ using the Book-Crossing dataset. As shown in Figure 2, as the error rate $\alpha$ increases, the performance across different metrics (MRR, Recall, NDCG) as well as the average set size across all models decreases. This decreasing trend demonstrates the framework's ability to generate valid prediction sets that adapt to the error rate $\alpha$.

We further evaluate the effect of varying confidence $\delta \in [0.05, 0.10, 0.15, 0.20, 0.25]$ on performance and average set sizes under fixed risk thresholds ($\alpha = 0.07$) using the Last.fm dataset in Figure 3. In general, all the models show a decreasing trend, validating the effectiveness of the framework. This is because relaxing confidence in risk constraints makes predictions less conservative, thereby reducing the number of items included in the recommendation set. Interestingly, performance and set sizes show a smaller decline for $\delta$ compared to $\alpha$, since $\delta$ controls only the confidence with which the risk constraint must hold i.e., the probability mass in the extreme tail, whereas $\alpha$ sets the risk level itself.

We also perform a grid study of the change-point parameters $\beta$ (shift sensitivity) and $\gamma$ (segment-length prior) on Book-Crossing dataset while holding all other settings fixed. Table 11 reports average

Table 5: Performance comparisons with base models ( NeuMF, CASER, SASRec and FMLP-Rec ) and user preference aware baselines ( TiSASRec, CDR and Oracle4Rec ) on **Taobao and MovieLens Datasets** using metrics ( MRR, Recall, NDCG ). For SURE, $\alpha$ and $\delta$ are set empirically as 0.05, respectively. Bold indicates the best result, and underline indicates the second best.

| Method | Taobao | | | MovieLens | | |
|---|---|---|---|---|---|---|
| | MRR ↑ | Recall ↑ | NDCG ↑ | MRR ↑ | Recall ↑ | NDCG ↑ |
| Model Ceiling@25(NeuMF) | 0.336 | 0.625 | 0.349 | 0.392 | 0.784 | 0.415 |
| NeuMF | 0.275 | 0.556 | 0.289 | 0.342 | 0.721 | 0.358 |
| NeuMF + SURE (Ours) | 0.292 | 0.587 | 0.298 | 0.356 | 0.739 | 0.368 |
| Model Ceiling@25(CASER) | 0.381 | 0.645 | 0.391 | 0.434 | 0.831 | 0.445 |
| CASER | 0.320 | 0.589 | 0.338 | 0.381 | 0.775 | 0.389 |
| CASER + SURE (Ours) | 0.343 | 0.612 | 0.350 | 0.391 | 0.798 | 0.395 |
| Model Ceiling@25(SASRec) | 0.395 | 0.663 | 0.408 | 0.458 | 0.854 | 0.469 |
| SASRec | 0.337 | 0.605 | 0.338 | 0.395 | 0.795 | 0.405 |
| SASRec + SURE (Ours) | 0.353 | 0.625 | 0.359 | 0.413 | 0.807 | 0.423 |
| Model Ceiling@25(FMLP-Rec) | 0.412 | 0.685 | 0.421 | 0.474 | 0.886 | 0.493 |
| FMLP-Rec | 0.363 | 0.612 | 0.361 | 0.405 | 0.811 | 0.415 |
| FMLP-Rec + SURE (Ours) | **0.373** | **0.649** | **0.385** | **0.435** | **0.851** | **0.454** |
| User Preference-Aware Models | | | | | | |
| TiSASRec | 0.348 | 0.610 | 0.353 | 0.402 | 0.802 | 0.412 |
| CDR | 0.339 | 0.609 | 0.351 | 0.399 | 0.795 | 0.405 |
| Oracle4Rec | 0.363 | 0.615 | 0.363 | 0.411 | 0.835 | 0.419 |

set size / coverage. We observe a consistent trade-off: larger $\beta$ or smaller $\gamma$ makes the detector more responsive, yielding slightly larger sets with improved coverage; the reverse favors tighter sets but risks transient under-coverage. In practice, we set $\beta=0.7$, $\gamma=1.1$ as a balanced choice across datasets. Finally, we vary the number of bootstrapped experts $L \in \{5, 10, 20\}$ and observe that SURE's set size and coverage are stable (Table 12). This empirical insensitivity is consistent with Theorem 5.1, which implies only a $\mathcal{O}(\sqrt{\ln L})$ growth term in the ensemble set size bound.

Overall, this parameter analysis guides real-world applications in balancing performance and recommendation set compactness with confidence guarantees.

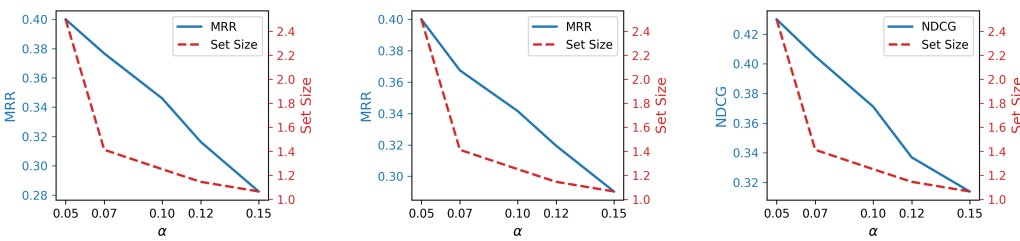

Figure 2: Performance analysis on the **Book-Crossing** dataset for varying $\alpha \in$ 0.05, 0.07, 0.10, 0.12, 0.15 with fixed $\delta = 0.05$, shown in terms of recommendation metrics and prediction set size.

Table 6: Performance comparisons with base models ( NeuMF, CASER, SASRec and FMLP-Rec ) and user preference aware baselines ( TiSASRec, CDR and Oracle4Rec ) on **Gowalla** using metrics ( MRR, Recall, NDCG ). For SURE, $\alpha$ and $\delta$ are set empirically as 0.05, respectively. Bold indicates the best result, and underline indicates the second best.

| Method | Gowalla | | |
|---|---|---|---|
| | MRR ↑ | Recall ↑ | NDCG ↑ |
| Model Ceiling@25(NeuMF) | 0.327 | 0.618 | 0.334 |
| NeuMF | 0.286 | 0.565 | 0.289 |
| NeuMF + SURE (Ours) | 0.291 | 0.577 | 0.309 |
| Model Ceiling@25(CASER) | 0.376 | 0.643 | 0.384 |
| CASER | 0.322 | 0.589 | 0.336 |
| CASER + SURE (Ours) | 0.334 | 0.602 | 0.343 |
| Model Ceiling@25(SASRec) | 0.385 | 0.667 | 0.394 |
| SASRec | 0.332 | 0.599 | 0.349 |
| SASRec + SURE (Ours) | 0.344 | 0.612 | 0.359 |
| Model Ceiling@25(FMLP-Rec) | 0.406 | 0.679 | 0.413 |
| FMLP-Rec | 0.342 | 0.605 | 0.355 |
| FMLP-Rec + SURE (Ours) | **0.359** | **0.632** | **0.364** |
| **User Preference-Aware Models** | | | |
| TiSASRec | 0.339 | 0.601 | 0.350 |
| CDR | 0.333 | 0.595 | 0.349 |
| Oracle4Rec | 0.343 | 0.609 | 0.360 |

Table 7: Comparison in terms in terms of coverage and average prediction set size with conformal baselines (Split Conformal, EnbPI and Online Conformal) evaluated on four base recommenders (NeuMF, CASER, SASRec, and FMLP-Rec) using the **Last.fM** dataset. The error rate is set as $\alpha = 0.10$. Bold indicates the best result, underline indicates the second best.

| Base Model | Coverage ↑ | | | | Set Size ↓ | | | |
|---|---|---|---|---|---|---|---|---|
| | Split | EnbPI | Online | SURE (Ours) | Split | EnbPI | Online | SURE (Ours) |
| NeuMF | 0.833 | 0.858 | 0.881 | 0.901 | 41 | 42 | 42 | 43 |
| CASER | 0.835 | 0.868 | 0.884 | 0.903 | 40 | 42 | 43 | 41 |
| SASRec | 0.849 | 0.870 | 0.889 | 0.905 | 40 | 41 | 42 | 40 |
| FMLP-Rec | 0.855 | 0.873 | 0.899 | **0.907** | 40 | 40 | 40 | **39** |

Table 8: Comparison in terms in terms of coverage and average prediction set size with conformal baselines (Split Conformal, EnbPI and Online Conformal) evaluated on four base recommenders (NeuMF, CASER, SASRec, and FMLP-Rec) using the **Taobao** dataset. The error rate is set as $\alpha = 0.10$. Bold indicates the best result, underline indicates the second best.

| Base Model | Coverage ↑ | | | | Set Size ↓ | | | |
|---|---|---|---|---|---|---|---|---|
| | Split | EnbPI | Online | SURE (Ours) | Split | EnbPI | Online | SURE (Ours) |
| NeuMF | 0.828 | 0.859 | 0.880 | 0.901 | 42 | 43 | 44 | 44 |
| CASER | 0.835 | 0.862 | 0.881 | 0.903 | 42 | 43 | 42 | 42 |
| SASRec | 0.836 | 0.871 | 0.900 | 0.909 | 41 | 42 | 42 | 41 |
| FMLP-Rec | 0.838 | 0.879 | 0.901 | **0.911** | 41 | 41 | 41 | **40** |

Table 9: Comparison in terms in terms of coverage and average prediction set size with conformal baselines (Split Conformal, EnbPI and Online Conformal) evaluated on four base recommenders (NeuMF, CASER, SASRec, and FMLP-Rec) using the **MovieLens** dataset. The error rate is set as $\alpha = 0.10$. Bold indicates the best result, underline indicates the second best.

| Base Model | Coverage ↑ | | | | Set Size ↓ | | | |
|---|---|---|---|---|---|---|---|---|
| | Split | EnbPI | Online | SURE (Ours) | Split | EnbPI | Online | SURE (Ours) |
| NeuMF | 0.851 | 0.859 | 0.862 | 0.901 | 39 | 40 | 40 | 39 |
| CASER | 0.861 | 0.878 | 0.872 | 0.901 | 39 | 40 | 40 | 38 |
| SASRec | 0.867 | 0.881 | 0.891 | **0.902** | 38 | 38 | 39 | 36 |
| FMLP-Rec | 0.871 | 0.889 | 0.901 | 0.901 | 38 | 37 | 38 | **35** |

Table 10: Comparison in terms in terms of coverage and average prediction set size with conformal baselines (Split Conformal, EnbPI and Online Conformal) evaluated on four base recommenders (NeuMF, CASER, SASRec, and FMLP-Rec) using the **Gowalla** dataset. The error rate is set as $\alpha = 0.10$. Bold indicates the best result, underline indicates the second best.

| Base Model | Coverage ↑ | | | | Set Size ↓ | | | |
|---|---|---|---|---|---|---|---|---|
| | Split | EnbPI | Online | SURE (Ours) | Split | EnbPI | Online | SURE (Ours) |
| NeuMF | 0.829 | 0.851 | 0.871 | 0.901 | 43 | 43 | 44 | 44 |
| CASER | 0.831 | 0.860 | 0.883 | 0.902 | 43 | 42 | 44 | 43 |
| SASRec | 0.837 | 0.870 | 0.895 | 0.901 | 43 | 42 | 43 | 42 |
| FMLP-Rec | 0.842 | 0.875 | 0.900 | **0.905** | 42 | 47 | 43 | **41** |

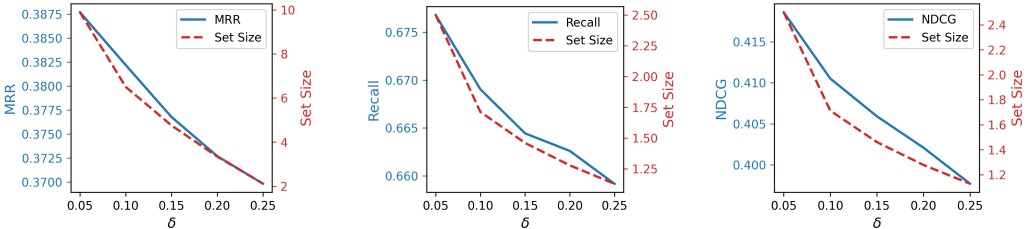

Figure 3: Performance analysis on the **Last.fm** dataset for varying $\delta \in 0.05, 0.1, 00.15, 0.20, 0.255$ with fixed $\alpha = 0.07$, shown in terms of recommendation metrics and prediction set size.

### A.7.4 ABLATION STUDY

To evaluate the effect of the two detection components in SURE, we perform an ablation study by selectively removing each loss-based shift term. We follow the same experimental protocol as described in Section A.6.2, with the error rate fixed at $\alpha = 0.1$ and confidence level $\delta = 0.05$. We report results on the Book-Crossing dataset with the SASRec backbone, and analyze the performance in terms of a) *validity*: measured as realized coverage against the error rate, b) *compactness*: measured in terms of the average set size, and c) *robustness*: which is measured in terms of the recommendation set volatility across the time stamps. We define the robustness parameter $\chi$ as:

$$\chi = \frac{1}{T-1} \sum_{t=2}^{T} \frac{\left| \mathcal{C}_{\boldsymbol{\lambda}^t}^{\mathrm{agg}} \Delta \mathcal{C}_{\boldsymbol{\lambda}^{t-1}}^{\mathrm{agg}} \right|}{\left| \mathcal{C}_{\boldsymbol{\lambda}^{t-1}}^{\mathrm{agg}} \right|},$$

where $\Delta$ denotes the difference between consecutive aggregated prediction sets.

We consider he following cases: (1) **w/o** $d_\ell^{\mathrm{ldd}}$, where only the concept-sensitive divergence $d_\ell^{\mathrm{con}}$ is retained; and (2) **w/o** $d_{\mathrm{con}}$, where only the loss discrepency distance $d_{\mathrm{ldd}}$ is retained. The results are denoted in Table 13. The results lead to following key observations:

Table 11: Set size (left) and coverage (right) for different $\gamma$ and $\beta$ on **Book-Crossing** Dataset.

| $\gamma \downarrow$ / $\beta \rightarrow$ | 0.5 | 0.7 | 1.0 |
|---|---|---|---|
| 0.9 | 44.6 / 0.920 | 45.6 / 0.924 | 46.5 / 0.930 |
| 1.1 | 42.1 / 0.895 | 42.8 / 0.908 | 43.5 / 0.912 |
| 1.3 | 41.2 / 0.889 | 42.2 / 0.892 | 43.1 / 0.901 |

Table 12: Robustness to ensemble size $L$ on **Book-Crossing** Dataset (set size / coverage).

| $L$ | 5 | 10 | 20 |
|---|---|---|---|
| set size / coverage | 42.5 / 0.906 | 42.9 / 0.908 | 43.5 / 0.908 |

- Firstly, removing $d_\ell^{\text{ldd}}$ substantially reduces validity. The coverage drops below the target $\alpha$. As a result, the framework tries to compensate by inflating the prediction sets. This is because, without the loss-discrepancy term, the detector becomes insensitive to uniform increases in difficulty across models. In such cases, shifts that affect all experts simultaneously go undetected, and calibration lags behind, leading to systematic under-coverage.

- Secondly, removing $d_\ell^{\text{con}}$ primarily degrades robustness. Although coverage remains close to the target and the average set size looks competitive, the volatility $\chi$ nearly doubles. This indicates unstable calibration as the threshold $\lambda$ fluctuates sharply in response to transient expert disagreements, even when the underlying distribution is relatively stable. In practice, this results in inconsistent recommendation sets from one time step to the next, potentially harming user trust.

- Finally, the full SURE framework, by jointly utilizing both the loss-discrepancy and the concept-sensitive terms, balances the strengths of each detector. The loss-discrepancy term guards against systematic difficulty shifts, while the concept-sensitive term dampens volatility caused by transient expert fluctuations. Their combination ensures that coverage stays close to the nominal target (validity), prediction sets remain as small as possible without sacrificing risk guarantees (efficiency), and threshold updates evolve smoothly over time (robustness).

These results show that each component is complementary and addresses a distinct failure mode, and together they form a balanced and reliable detector of preference shifts. Hence, both signals are indispensable for achieving stable uncertainty-aware recommendations under non-stationary user behavior.

### A.8 Intuition of Adaptive Dynamics in SURE

To provide an intuitive understanding of the SURE framework's adaptive capability, we visualize the internal dynamics of the DAUO algorithm during a user session based on interactions from the Taobao dataset, designed to illustrate a sequence of preference shifts. Figure 4 describes how the three key variables evolve: the rolling risk, the calibration threshold ($\lambda$), and the prediction set size.

Figure 4 illustrates the clear causal sequence of the adaptation loop. Initially, stable user behavior allows for a high threshold ($\lambda \approx 0.62$) and compact set size. A sudden preference shift degrades the ranking quality, causing a risk spike. The DAUO update rule (Eq. 15) counters this by lowering $\lambda$ (Middle Panel), which accordingly expands the prediction set (Bottom Panel) to restore coverage. Notably, the set size stabilizes at a higher level rather than returning to baseline because the underlying backbone model remains frozen. SURE correctly identifies that the frozen model is now less accurate for the new user preference and permanently maintains a larger safety margin to ensure continued risk control.

### A.9 Discussion

Our framework SURE reframes sequential recommendation as an uncertainty-aware prediction set problem that (1) hedges an ensemble of bootstrapped recommenders through Hedge weighting with

Table 13: Ablation of detection components on **Book-Crossing** Dataset

| Variant | Coverage ↑ | Avg set size ↓ | Volatility $\chi$ ↓ |
|---|---|---|---|
| SURE | 0.908 | 42.8 | 0.12 |
| w/o $d_\ell^{\text{ldd}}$ | 0.872 | 44.7 | 0.10 |
| w/o $d_\ell^{\text{con}}$ | 0.907 | 43.1 | 0.23 |

adaptive conformal thresholds, (2) detects user-specific preference shifts without any heuristically chosen window lengths utilizing a Bayesian changepoint detection model, and (3) provides sample guarantees that both the expected set size and the utility-based risk stay near-optimal under non-stationary preferences. Our claims are empirically supported as SURE consistently outperforms base recommender models and preference-aware recommender baselines on various recommendation metrics while maintaining tight and valid $(1 - \alpha)$ coverage across five public datasets. It does so without adding any significant training time, hence it can be expanded to recent popular generative models (Rajput et al., 2023; Zhai et al., 2024; Deng et al., 2025; Han et al., 2025). It is also robust in addressing broader concerns raised in the recommendations. Because thresholds and ensemble weights are updated externally with respect to a platform-defined utility function $U_{metric}$, the framework can incorporate fairness- or diversity-aware objectives directly. For example, $U_{metric}$ can be defined to penalize concentration or unsafe content, or combined with exposure caps and pre-filters; the coverage guarantees then hold with respect to this modified $U_{metric}$, requiring no change to the theory. This flexibility ensures resilience to issues such as filter bubbles or echo chambers. Different fairness definitions across user groups is also supported by the mechanism. Since thresholds and Hedge weights are updated externally, calibration can be performed separately for groups (e.g., by demographics, region, or activity level). Replacing $|\mathcal{U}|$ with $|\mathcal{U}_g|$ yields valid guarantees for each group independently, preserving equitable coverage across heterogeneous populations. Users in smaller or sparser cohorts may see slightly larger average set sizes due to finite-sample slack, but validity is preserved as shown in Theorem 5.1 and Theorem 5.2.

SURE does face the finite-sample effect. While the smaller calibration size continues ensuring the validity in a dynamic environment, it may lead to more conservative prediction sets as shown in our theoretical results. Also, as commonly seen in conformal strategies, SURE can only be as good as the confidence scores it calibrates. If a backbone recommender produces poorly ranked logits with poorly calibrated backbones (NeuMF), SURE's sets are ∼15% larger than with stronger models (FMLP-Rec). We aim to address these challenges in future work. Overall, our work bridges the gap between sequential recommender systems' lack of reliability in adaptive environments with changing user preferences, which is a pragmatic step towards inspiring future research in trustworthy recommendation systems.

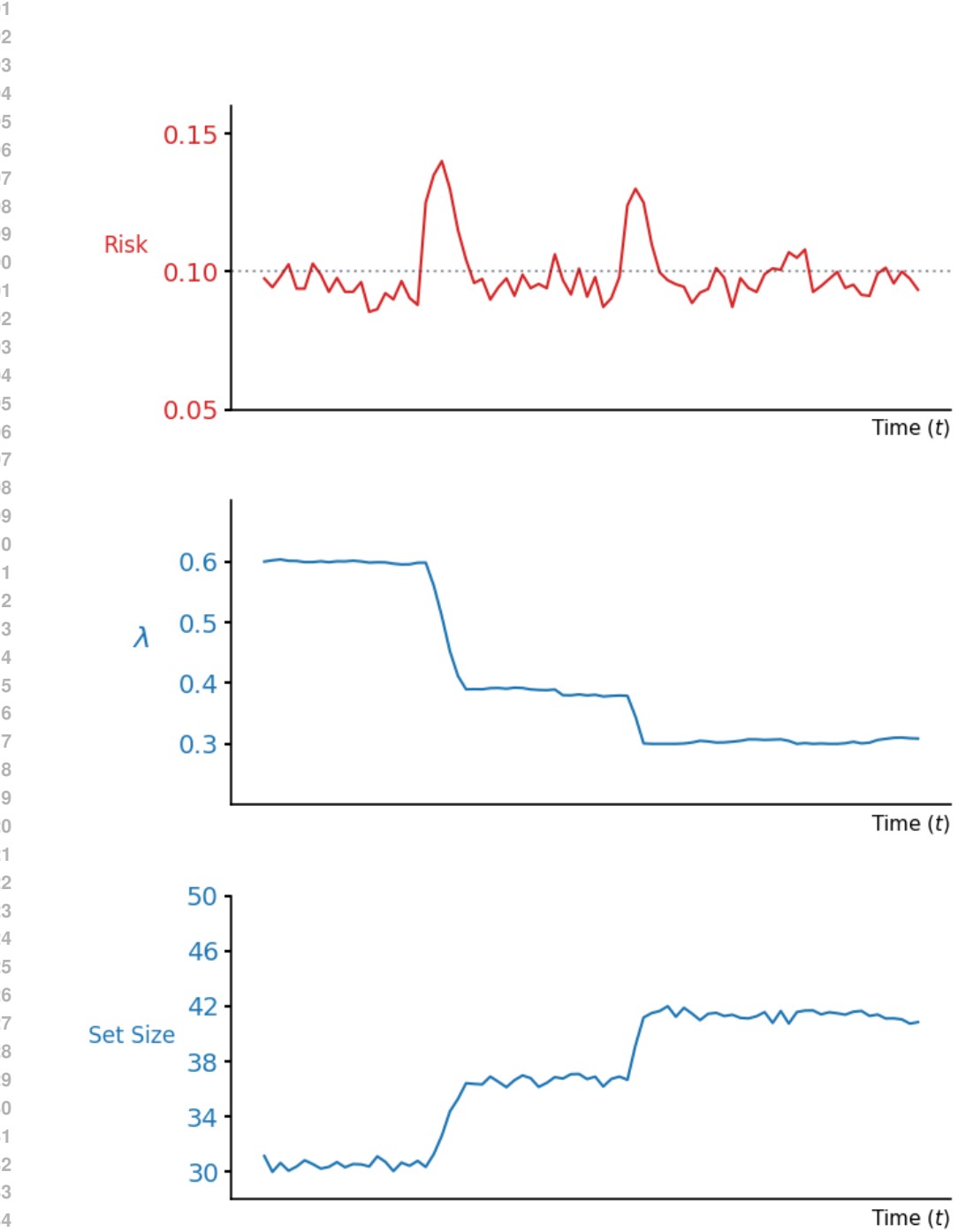

Figure 4: **Dynamic Adaptation of SURE under Preference Shift.** (Top) The rolling risk spikes above the target $\alpha = 0.10$, indicating preference shifts. (Middle) The calibration threshold $\lambda^t$ reacts immediately by lowering (Eq. 15) to loosen constraints. (Bottom) The prediction set size accordingly increases, confirming the framework's ability to actively detect and correct for preference shift in real-time.

