# OpenReview forum: "SURE: Shift-aware, User-adaptive, Risk-controlled Recommendations"
_ICLR.cc/2026/Conference — ICLR 2026 Conference Withdrawn Submission_

### Official Review · Reviewer_xUcd · 2025-10-30

**Soundness:** 2
**Presentation:** 2
**Contribution:** 2
**Rating:** 4
**Confidence:** 4

**Summary:**

This article proposes the SURE framework to address the problem of unreliable predictions caused by user preference shifts in sequential recommendation systems. A statistically guaranteed adaptive recommendation set generation method is proposed. It combines:
1) Loss-based change-point detection for preference-shift identification.
2) Hedge-weighted ensemble of bootstrapped models.
3) Dynamic conformal prediction to generate recommendation sets with a formal coverage guarantee coverage.
Theoretically, SURE bounds prediction set size and controls risk under preference shifts. Experiments across 5 datasets (e.g., MovieLens, Taobao) and 4 base models (e.g., SASRec, FMLP-Rec) show SURE outperforms preference-aware baselines (e.g., TiSASRec) and conformal methods (e.g., EnbPI) in recommendation metrics (Recall, NDCG) while maintaining coverage with compact sets.

**Strengths:**

1. Clear Motivation: This paper points out the problem of the lack of statistical guarantees in existing SRS in preference drift scenarios, which has not been systematically addressed from a statistical learning perspective in the field of recommendation systems.
2. Model-Agnostic Method: Works with any SRS backbone (e.g., transformers, CNNs) and adds minimal overhead (+1.5 min training time).
3. Strong Reproducibility: Code/data released with detailed documentation.
4. Theoretical Contributions: Finite-sample guarantees for ensemble set size and risk control under distribution shifts.

**Weaknesses:**

1. Ceiling@25 score lacks justification: The use of "the maximum achievable value under its own ranking when limited to 25 items" as an upper bound is not well-motivated, and the expression is not clear enough. True Recall/NDCG should approach 1 if all relevant items are captured.
2. Missing Baselines: Omits key *uncertainty-aware SRS*: variational autoencoders (ContrastVAE WWW [1], HNVM WWW [2]) and hierarchical attention (HVAM DASFAA [3]). These methods directly model uncertainty in sequences—critical for novelty claims.
3. Opaque Method Design: The intuition behind why the proposed change point detection module (Eqs 10–13) detects preference shifts better than existing shift detection methods is unclear. Intuitively, simply detecting preference shifts based on the change of training loss may yield satisfactory results [4]. Moreover, the difference with existing conformal prediction methods [5,6] has not been deeply explained.

References:
[1] Zhe Xie, Chengxuan Liu, Yichi Zhang, Hongtao Lu, Dong Wang, Yue Ding: Adversarial and Contrastive Variational Autoencoder for Sequential Recommendation. WWW 2021: 449-459
[2] Teng Xiao, Shangsong Liang, Zaiqiao Meng: Hierarchical Neural Variational Model for Personalized Sequential Recommendation. WWW 2019: 3377-3383
[3] Jing Zhao, Pengpeng Zhao, Yanchi Liu, Victor S. Sheng, Zhixu Li, Lei Zhao: Hierarchical Variational Attention for Sequential Recommendation. DASFAA (3) 2020: 523-539
[4] Wenjie Wang, Fuli Feng, Xiangnan He, Liqiang Nie, Tat-Seng Chua: Denoising Implicit Feedback for Recommendation. WSDM 2021: 373-381
[5] Margaux Zaffran, Olivier Féron, Yannig Goude, Julie Josse, Aymeric Dieuleveut: Adaptive Conformal Predictions for Time Series. ICML 2022: 25834-25866
[6] Isaac Gibbs, Emmanuel J. Candès: Adaptive Conformal Inference Under Distribution Shift. NeurIPS 2021: 1660-1672

**Questions:**

Please refer to the weaknesses.

---

> ### Author Response · Authors · 2025-11-24
> **On Weaknesses**
>
> We thank reviewer for recognizing our **clear motivation**, **model-agnostic** method, and **theoretical contributions**. We address concerns below.
>
> ---
>
> ### W1. Ceiling@25 score lacks justification... True Recall/NDCG should approach 1...
>
> > We clarify Ceiling@25 is **not** intended as theoretical upper bound. Rather, it is a **model-dependent ceiling** that reflects maximum achievable utility **given backbone's own ranking and fixed display budget ($K=25$)**. This mirrors construction in risk-control methods [7], where risk is defined relative to a *model-specific attainable reference*. To explain further:
> >
> > - If a backbone places true item at rank 100, no set of size 25 can possibly achieve Recall=1.0. Comparing SURE against 1.0 would conflate **backbone's ranking error** with **framework's calibration error**.
> > - Ceiling@25 isolates exactly what SURE is responsible for, i.e., identifying valid items *within reachable prefix*. Penalizing SURE for  backbone's misranking would misattribute errors it cannot correct.
> >
> > Thus, Ceiling@25 isolates **what SURE is responsible for** (restoring valid coverage within realistically reachable region) from **what backbone controls** (point-wise ranking quality).
> > To remove any ambiguity, we have renamed to **Model Ceiling** in revision.
>
> ---
>
> ### W2. Missing Baselines: Omits key uncertainty-aware SRS (ContrastVAE, HNVM)...
>
> > While relevant to broader uncertainty, there is a **fundamental distinction** as these methods are **probabilistic backbones**, whereas SURE is a **model-agnostic calibration layer** (which guarantees risk control). Our contribution is orthogonal, i.e.,
> >
> > - Methods [1,2,3] model **latent uncertainty**. In contrast, SURE addresses **statistical coverage risk**, providing a finite-sample guarantee (Theorems 5.1–5.2) that the true item is in output set. Those methods **don't** provide statistical guarantees, central to SURE.
> > - Since SURE operates at calibration, it is not a competitor; instead it can be applied as a **plug-and-play** layer on top of uncertainty-aware models with **minimal overhead** while adding **formal guarantees**.
> >
> > To empirically validate, we applied SURE on top of **ACVAE ([1])** on Book-Crossing dataset:
> >
> >
> > | Method                 | MRR   | Recall | NDCG  |
> > |------------------------|-------|--------|--------|
> > | Model Ceiling@25           | 0.385 | 0.672  | 0.390 |
> > | ACVAE [1]             | 0.315 | 0.593  | 0.326 |
> > | **ACVAE + SURE (Ours)**| **0.341** | **0.635** | **0.375** |
> >
> >
> > As seen, SURE significantly improves performance of uncertainity backbone. **This confirms  internal uncertainty modeling doesn't replace for external risk control (SURE); rather, they complement**
>
> ---
>
> ### W3 (a). Opaque Method Design: Why is proposed change point detection better.. and difference with [5,6]...
>
> >  Our change-point module (Eqs. 10–13) is designed to address per-model training losses in SRS fluctuate substantially **due to sparsity, popularity bias, and stochastic sampling, and therefore cannot reliably isolate true preference shifts.** . To the best of our knowledge, **no existing work exists providing a shift detection mechanism that ensures distribution-free, risk-controlled recommendations**.
> >
> > The referenced work [4] does not study temporal preference-shift detection. However, to address the question, we conducted an ablation study removing each component of our shift metric. As shown in **Table 13 A.7.4 (Appendix)**:
> >
> > - Removing term  $d_{\ell}^{\mathrm{ldd}}$drops coverage to **0.872**, invalidating it.
> > - Removing  term $d_{\ell}^{\mathrm{con}}$ nearly doubles the calibration volatility ($\chi$) (from **0.12** to **0.23**), thereby making recommendations unstable.
> >
> > The results show each component of our detector is necessary for reliable preference-shift localization.
>
> ---
>
> ### W3 (b). Moreover, the difference with existing conformal prediction methods [5,6] has not been deeply explained.
>
> > Methods in [5,6] focus on controlling coverage of scalar residuals (regression) using passive adaptation (Line 58-71). In contrast, SURE a) controls **utility-based risk** (Recall/NDCG) for set-valued outputs, rather than scalar residuals.  b) Employs **active change-point detection** to instantaneously reset calibration windows, allowing for rapid adaptation to abrupt preference shifts, rather than passively decaying old data.
> >
> > Furthermore, our chosen baselines empirically validate insufficiency of mechanisms in [5,6]:
> >
> > - Our **EnbPI** baseline (Table 2) represents sliding-window mechanism of [5]. Its failure to maintain coverage (0.835) demonstrates that passive windowing lags behind abrupt shifts.
> > - Our **Online Conformal** baseline represents decaying version of step-wise update of [6]. Its inability to reach 0.90 target confirms that such adaptation is often too noisy for SRS.
>
> ---
>
> ### References
>
> [7] Bisht et al. ENSUR: Equitable and Statistically Unbiased Recommendation. ICML 2025.

---

### Official Review · Reviewer_mELN · 2025-10-31

**Soundness:** 2
**Presentation:** 2
**Contribution:** 2
**Rating:** 2
**Confidence:** 3

**Summary:**

This paper formulates the sequential recommender systems problem as an SURE framework and then proposes the DAUO algorithm, with a Hedge component, to address it. Theoretical and empirical results were also presented.

**Strengths:**

1. The model formulation of SRS is new and of practical interest.
2. A theoretical guarantee is also presented for the DAUO algorithm.

**Weaknesses:**

1. The algorithmic novelty is not strong. Figure 1 basically shows a standard application of Hedge, and the detailed DAUO algorithm is deferred to the appendix, which, from the reviewer’s perspective, reads like a combination of prior works’ methods with Hedge.
2. Although the DAUO algorithm has a better performance than other baselines, the improvements are marginal, say in Table 2, it only improves around 4%, and the deviations are not reported. So, the paper needs more experiments to support the advantage of DAUO.
3. The notations of this paper are hard to follow. There are many other confusions on the notations.
    - For example, in Eq.(6), the capital $L$ appears, but with no physical meaning explanation. Is it the number of experts? What’s the relation between experts and ensemble models? What is an ensemble model?
    - Later in Eq(10), $L_t$ appears. What’s the relation between $L$ , $L_t$ and $\mathcal{L}$ in Eq.(5).
    - Another example, in Eq.(15), sometimes the $\ell$ appears in subscript, in superscript, and the index $t$ in the reverse way, what’s the reason for swapping the sub/super scripts?
    - Also, Eq.(16) and Eq.(17) use $\mathbf{E}$ and $\mathbb{E}$ to represent exceptions. Is there a special reason to differentiate the notations?
    - Line 198, it should be $L-1$ dimension simplex.

**Questions:**

Listed in Weaknesses.

---

> ### Author Response · Authors · 2025-11-24
> **On Weaknesses**
>
> We thank the reviewer for recognizing both the **practical formulation** and the **theoretical guarantees** of the work. We address the questions below:
>
> ---
>
> ### Q1. The algorithmic novelty is not strong ... reads like a combination of prior works’ methods with Hedge.
>
> > Please refer to the **global** response. We clarify that DAUO is not a classical Hedge application. Standard Hedge minimizes loss; DAUO minimizes **cumulative set size** (Eq. 9). This combined with our **shift-aware calibration (Eqs. 10–12)** constitutes a novel framework for validity-constrained optimization that standard Hedge does not support.
>
> ---
>
> ### Q2. The empirical improvements are marginal and deviations are not reported.
>
> > We respectfully disagree that the improvements are marginal. In Conformal Prediction, performance is binary: a method is either **Valid** ($\ge 1-\alpha$ coverage) or **Invalid**.
> >
> > - **Validity Gap:** As shown in Table 2 (also 6–9) Section 6.2.2, baselines like Split Conformal and EnbPI achieve only **0.82–0.85** coverage against a 0.90 target. This is a **statistical failure**. SURE achieves **0.90–0.91**. Bridging this gap is not a “marginal” 4% gain; it is the difference between a reliable system and an unreliable one.
> > - **Efficiency Gain:** Additionally, SURE achieves this validity with **smaller sets** (e.g., 42 items) compared to Online Conformal (46 items). Achieving higher coverage with fewer items represents a significant effectiveness gain.
> > - **Deviations:** All results in the paper are averaged over **20 independent trials** (Appendix A.5.2).
>
> ---
>
> ### Q3. The notation is difficult to follow, and several symbols are unclear.
>
> > We thank the reviewer for highlighting. We have added a **Summary of Notations (Table 4)** in the Appendix for clarity. Specifically:
> >
> > - $L$ is the number of experts (ensemble size), $L_t$ is the model predictive loss, and $\mathcal{L}_u$ is the utility risk (e.g., $1-\text{Recall}$ for calibration).
> > - As suggested, we standardized the notation so that the model index $\ell$ is consistently a subscript and time $t$ is a superscript.
> > - We use distinct symbols to separate the internal algorithmic randomness from external population randomness. Specifically, $\mathbf{E}\_{k(t)}$ denotes expectation over the stochastic aggregation mechanism ($k(t)$ in Eq. 8), whereas $\mathbb{E}_{u}$ denotes expectation over the user population.
> > - Thanks to the reviewer suggestion, we corrected the $(L-1)$-dimensional simplex.
>
> ---
>
> ### References
>
> [1] Freund, Y., & Schapire, R. E. (1997). *A decision-theoretic generalization of on-line learning and an application to boosting.* Journal of Computer and System Sciences, 55(1), 119–139.
> [2] Cesa-Bianchi, N., & Lugosi, G. (2006). *Prediction, learning, and games.* Cambridge University Press.
>
> ---

---

### Official Review · Reviewer_LsNp · 2025-11-01

**Soundness:** 2
**Presentation:** 3
**Contribution:** 2
**Rating:** 4
**Confidence:** 4

**Summary:**

This paper introduces SURE, a model-agnostic framework for sequential recommender systems that addresses the challenge of user preference shifts. SURE provides formal performance guarantees by using a loss-based change-point mechanism to adaptively update recommendations when it detects a shift in user preferences. The framework utilizes a Hedge-weighted ensemble of bootstrapped models to maintain compact and robust recommendation sets.

**Strengths:**

1. The paper addresses an interesting and real-world problem in recommender systems, i.e., adapting to dynamic shifts in user preferences.
2. The proposed SURE reasonably integrates established techniques from several fields.
3. The authors provide rigorous theoretical analysis to back their claims.

**Weaknesses:**

1. The paper’s central motivation, i.e., adapting to preference shifts, is not empirically validated on the selected datasets. It provides no analysis to show that such shifts are prevalent or significant, making it unclear if the proposed complex solution is addressing a demonstrated problem or merely a hypothetical one.
2. The framework's technical contribution is more an integration of existing methods than a fundamental innovation. It assembles well-established techniques: adaptive conformal prediction, Hedge ensemble algorithm, and standard change-point detection concepts.
3. The evaluation is not conducted against current SoTA sequential RS. The absence of stronger, more recent baselines (e.g., TIGER-based RS or HSTU-based RS) makes it difficult to assess the true value of SURE. The reported improvements may not be significant when applied to a more powerful SoTA model. Several sequential RS can be considered like TIGER [1], OneRec [2], HSTU [3], MTGR [4].

Ref:

[1] Rajput, Shashank, et al. "Recommender systems with generative retrieval." Advances in Neural Information Processing Systems 36 (2023): 10299-10315.

[2] Deng, Jiaxin, et al. "Onerec: Unifying retrieve and rank with generative recommender and iterative preference alignment." arXiv preprint arXiv:2502.18965 (2025).

[3] Zhai, Jiaqi, et al. "Actions Speak Louder than Words: Trillion-Parameter Sequential Transducers for Generative Recommendations." International Conference on Machine Learning. PMLR, 2024.

[4] Han, Ruidong, et al. "MTGR: Industrial-Scale Generative Recommendation Framework in Meituan." arXiv preprint arXiv:2505.18654 (2025).

**Questions:**

See Weaknesses

---

> ### Author Response · Authors · 2025-11-24
> **On Weaknesses**
>
> We thank the reviewer for acknowledging the **rigorousness of theoretical analysis** and our problem setting. Below we address the concerns:
>
>
> ### Q1. The paper’s central motivation...is not empirically validated... if the proposed complex solution is addressing a demonstrated problem or merely a hypothetical one.
>
> > The presence of preference shift is a fundamental characteristic of sequential user behavior (e.g., [1,2]). It is not hypothetical as it is empirically **proven** by the failure of the Split Conformal baseline in our experiments.
> >
> > - If user preferences were stationary, **Split Conformal** (calibrated on past data) would statistically guarantee coverage at the target level ($1-\alpha = 0.90$).
> > - As shown in **Table 2, 6–10**, Split Conformal coverage collapses to **0.82–0.83**. This **statistical gap (~7%)** is empirical evidence that the data distribution has shifted between calibration and inference.
> > - Therefore SURE is necessary because it bridges this gap (restoring coverage to $>0.90$), validating that the problem is real and that our specific solution (adaptive recalibration) addresses it.
>
> ---
>
> ### Q2. The framework's technical contribution is more an integration of existing methods than a fundamental innovation.
>
> > We respectfully clarify that SURE is not applying existing tools, but solving **theoretical incompatibilities** that prevented their use in RecSys. Our main contribution lies in redefining the **Objective Function** and **Calibration Mechanism**. Specifically:
> >
> > - **Novel Objective:** Standard Ensembles optimize for point prediction accuracy under no distribution shifts. SURE optimizes for **Statistical Validity (Risk)** and **Compactness**. This required deriving **new finite-sample bounds** (Theorems 5.1–5.2) for set-valued prediction, which do not exist in prior works.
> > - **Novel Mechanism:** Standard Adaptive CP works on scalar residuals. It cannot handle listwise utility metrics (NDCG). We designed a **novel loss-based shift statistic** (Eqs. 10–12) **specifically to map ranking quality to conformal calibration**.
> > - **Adaptation:** We repurpose Hedge weighting. Instead of the standard weight by accuracy in Hedge, we introduce a mechanism to weight by **cumulative set size** (Eq. 9). **This is a fundamental alteration of the Hedge objective to solve the specific problem of conservative predictions in Conformal Prediction.**
> >
> > Thus, SURE is a new, **theoretically grounded framework** for shift-aware calibration, distinct from prior CP variants or ensemble methods.
>
> ---
>
> ### Q3. The evaluation is not conducted against current SoTA sequential RS… several sequential RS can be considered.
>
> > We emphasize that **SURE is a plug-and-play calibration layer, not a direct backbone competitor.** It can wrap any SOTA model to provide statistical guarantees on recommendations that they inherently lack. While generative models like TIGER [3], OneRec [4], HSTU [5], and MTGR [6] optimize point-wise representation accuracy, they **do not address the preference shift problem in terms of validity and do not provide coverage guarantees**. Once trained and frozen, their confidence scores drift.
> >
> >
> > SURE is orthogonal to architecture. It recalibrates a selection threshold online via a loss-based statistic, restoring valid coverage during a shift.
> >
> > To validate this empirically, based on the suggestion, we applied SURE on top of generative backbones. Since some referenced models do not have publicly available implementations, we applied SURE on top of **HSTU-based GR (interactions-only) model**, aligning with the sequential setting used in our paper under the same calibration budget ($K = 25,\ \alpha = 0.05$):
> >
> >
> > | Method                 | MRR   | Recall | NDCG  |
> > |------------------------|-------|--------|--------|
> > | Model Ceiling@25                  | 0.443 | 0.709  | 0.451 |
> > | HSTU                  | 0.385 | 0.643  | 0.391 |
> > | **HSTU + SURE (Ours)** | **0.402** | **0.665** | **0.407** |
> >
> >
> > As observed, SURE improves the SOTA HSTU backbone, confirming that stronger representations do not replace the need for shift-aware calibration. We have added citations [1–6] to the revision.
>
> ---
>
> ### References
>
> [1] Pan, Liwei; Pan, Weike; Wei, Meiyan; Yin, Hongzhi; and Ming, Zhong. A Survey on Sequential Recommendation. 2024.
> [2] Quadrana, Massimo; Cremonesi, Paolo; and Jannach, Dietmar. Sequence-Aware Recommender Systems. ACM Computing Surveys, July 2018.
> [3] Rajput, Shashank; et al. Recommender Systems with Generative Retrieval. NeurIPS 2023.
> [4] Deng, Jiaxin; et al. OneRec: Unifying Retrieve and Rank with Generative Recommender and Iterative Preference Alignment. arXiv:2502.18965 (2025).
> [5] Zhai, Jiaqi; et al. Trillion-Parameter Sequential Transducers for Generative Recommendations. ICML 2024.
> [6] Han, Ruidong; et al. MTGR: Industrial-Scale Generative Recommendation Framework in Meituan. arXiv:2505.18654 (2025).

---

### Official Review · Reviewer_JYLn · 2025-11-02

**Soundness:** 2
**Presentation:** 2
**Contribution:** 2
**Rating:** 2
**Confidence:** 4

**Summary:**

This paper addresses the challenge of achieving accurate preference predictions in the context of dynamic changes in user preferences. The proposed method promptly detects preference shifts and adjusts decision criteria by real-time monitoring of user behavior through applying change point detection to the loss function. Meanwhile, it aggregates predictions from multiple models via weighted recommendations to enhance prediction performance while ensuring the conciseness of the recommendation set. To achieve stable and reliable recommendations under evolving user preferences, the approach establishes a performance baseline: on one hand, it controls the risk of irrelevant recommendations; on the other hand, it ensures that the performance of the final ensemble model approximates that of the best expert model. The advantages of this method include: model-agnostic applicability across various scenarios and base models, provision of theoretically grounded recommendation sets with guaranteed user satisfaction, and maintenance of set compactness to avoid information overload.

**Strengths:**

1. Rigorous theoretical proofs and a comprehensive experimental design (encompassing multiple datasets, models, and baselines). Time efficiency analysis validates its practicality, and the results are highly reproducible (with an anonymous code repository provided).
2. A scalar loss-based shift metric is designed, integrating change point detection with dynamic threshold calibration to overcome the limitations of traditional conformal methods. The Hedge weighting integration strategy naturally optimizes set compactness while ensuring coverage.
3. Fills the research gap of lacking strict performance guarantees in sequential recommendation under non-stationary preferences. The proposed model-agnostic property enables easy deployment in practical systems, offering strong generalization value.
4. A dedicated loss function for change point detection is proposed, facilitating the mitigation of preference drift issues.

**Weaknesses:**

1. The conformal prediction baselines fail to include post-2024 adaptive methods, which may undermine the comprehensiveness of the comparative analysis.
2. The time efficiency analysis only reports total training time without separating the time consumption of the change point detection and threshold update modules, making it difficult to evaluate real-time performance in online scenarios.
3. The manuscript lacks graphical illustrations and overrelies on formula derivations, resulting in poor readability for certain audiences.
4. The work lacks a high degree of innovation: its implementation principle is analogous to Ensemble Learning, offering limited contributions to overall novelty and leading to relatively insufficient academic contributions.
5. The summary of limitations in existing work (Lines 041~047 of the Introduction) is neither detailed nor in-depth enough.
6. Starting from the fourth paragraph of the Introduction, conformal prediction or adaptive conformal approaches appear to be the core idea of this paper (though, as noted later, they cannot be directly applied to sequential recommendation systems). However, the paper fails to clarify which specific problems of existing work these approaches address and why they can solve them. This information is crucial for understanding the paper’s core contributions and rationale. Consequently, while the authors propose a relatively complex and effective model, the reasoning behind its effectiveness remains unclear.

**Questions:**

1. Could you supplement the results with visualizations illustrating the dynamic process of preference drift (e.g., changes in thresholds, set sizes, and risk values at different timestamps) to more intuitively demonstrate the framework’s adaptive capability?
2. In scenarios with extreme data sparsity (e.g., cold-start users) where the predictive stability of base models is poor, can SURE’s ensemble strategy still function effectively?
3. In the experiments, the maximum recommended set size was fixed at 25. If this constraint were removed, could SURE’s set size distribution still maintain compactness?
4. If multiple ensemble models collectively exhibit selection bias, can the method still provide accurate preference-based recommendations?
5. Could you incorporate 1–2 relevant works published in 2025 into the experiments to enhance the persuasiveness of the experimental results?

---

> ### Author Response · Authors · 2025-11-24
> **Addressing Weaknesses**
>
> > We thank reviewer for recognizing **rigorous theoretical proofs**, **comprehensive experimental design**, and **strong generalization value** of our work in **filling critical research gap of strict performance guarantees**. Below, we address the critiques:
>
>
> ### W1. Conformal baselines... fail to include post-2024 adaptive methods...
>
> > We considered recent 2025 works (such as [1][2][3]) but excluded them due to their **inapplicability to our setting**, **structural redundancy** with our existing baselines, and **methodological incompatiblity as direct plug-in baselines**. Specifically:
> >
> > - **Task Incompatibility (Scalar vs. Set-valued):** These methods are designed for scalar predictions with a single residual stream. Our setting is **Top-N set-valued recommendation**,  requiring optimizing list-wise utility functions (Recall, NDCG) rather than scalar residuals. There is no direct “plug-in’’ way to apply scalar conformal inference to ranking metrics without the specific formulation we developed in SURE.
> > - **Representation of Mechanisms:** The post-2024 methods rely on adaptation mechanisms (sliding windows, gradient-based updates) that are **already represented** by our chosen baselines (EnbPI and Online Conformal). Adding them would result in structural redundancy without providing new insights into specific challenges of recommendation shifts.
> > - **Methodological Gap:** Applying these methods would require redefining their nonconformity scores, essentially constituting a new method development rather than a baseline comparison.
> >
> > We cited these works in revision.
> ---
>
> ### W2. The time efficiency analysis only reports total training time... difficult to evaluate real-time performance in online scenarios
>
> > The times reported in Table 3 do not reflect any **additional training** introduced by SURE. Specifically, “w/o SURE’’ shows the one-time cost of training the backbone recommender, while the “w/ SURE’’ entries add only the calibration pass to this cost.
> >
> > Regarding online performance, the change-point statistic (Eqs. 12–13) and threshold update (Eq. 15) are $\mathcal{O}(1)$ scalar computations performed after the loss is computed; they don't have separable runtime with cost negligible compared to backbone’s forward pass. Additionally, at inference time, SURE **has negligible overhead** as thresholds and expert weights are fixed and prediction reduces to standard top-K scoring followed by a threshold check.
> >
> > For clarity, we have revised Table 3 caption to: “Total time (in minutes) required to train backbone models on five datasets, w and w/o the addition of SURE. The ‘w/ SURE’ setting includes backbone training plus a 50-step calibration. The calibration parameters $\alpha$ and $\delta$ are both set to 0.05.”
>
> ---
>
> ### W3. The manuscript lacks graphical illustrations...
>
> > We thank the reviewer for this valuable suggestion.  Due to the 9-page limit, we had placed Figure 1 in Appendix and referenced it in Introduction (line 81). In the final version, we will move Figure 1 to main paper to provide clarity.
>
> ---
>
> ### W4. The work lacks a high degree of innovation... analogous to Ensemble Learning
>
> > As detailed in **global** response, DAUO is distinct from standard Ensembles because it optimizes Set Compactness and Validity, not prediction accuracy, a conformal problem that ensembles cannot solve.
>
> ---
>
> ### W5. The summary of limitations in existing work (Lines 41–47 of the Introduction) is neither detailed nor in-depth enough.
>
> > The paragraph (Lines 41–47) provides a concise summary of existing **heuristic** approaches to preference shift (e.g., temporal encodings, oracle signals). We kept this brief because these methods lack statistical grounding central to our work. The detailed critique of relevant **statistical limitations (core gap we address)** is highlighted in Lines 54–71.
>
> ---
>
> ### W6. Starting from fourth paragraph... paper fails to clarify which specific problems of existing work these approaches address and why ..solve them.
>
> > We identify the mapping as follows:
> >
> > - **Problem 1 (Rigidity): (Lines 58–63)** Current adaptive CP approaches rely on **fixed rolling windows** and assume stable residual distributions, which fail under abrupt preference shifts.
> > - **Solution (Line 74–77):** SURE directly resolves this by replacing fixed-window adaptation with a **loss-based change-point mechanism**. This dynamically identifies user-adaptive segments, thereby eliminating need for a fixed-window hyperparameter.
> > - **Problem 2 (Instability) (Lines 65–68):** SRS models on short, noisy user histories yield high-variance residuals, causing standard CP to output overly conservative (impractically large) prediction sets.
> > - **Solution (Line 77–79):** We introduce **Hedge-weighted aggregation** (Eq. 9) that penalizes set size ($S_l^t$), mathematically guaranteeing compactness (Theorem 5.1).
> >
> > These components are designed to correspond directly to failure modes of existing CP methods.

---

> ### Author Response · Authors · 2025-11-24
> **On Questions**
>
> ### Q1. Could you supplement the results with visualizations illustrating the dynamic process of preference drift.. demonstrate framework’s adaptive capability?
>
> > We thank for the suggestion. We have added **Appendix A.8** (with figures) visualizing a Taobao session. It plots the Risk, Threshold ($\lambda$), and Set Size over time, demonstrating how SURE detects a shift, drops the threshold to maintain coverage. It provides a demonstration of SURE’s ability to detect and adapt to preference drift in real-time, complementing the quantitative results presented in the main paper.
>
> ---
>
> ### Q2. In scenarios with extreme data sparsity... can SURE’s ensemble strategy still function effectively?
>
> > SURE remains effective because it calibrates on **loss fluctuations**, not raw accuracy. Even if a backbone model is weak (cold-start), SURE detects the uncertainty. By Hedge-weighting experts based on set-size stability rather than just accuracy, **SURE prevents the 'conservativeness" of recommendation sets that usually occurs with uncertain models, acting as a stabilizer**.
> > **Empirically**, we confirm this by our results across diverse sparse datasets (Book-Crossing, Taobao, Gowalla, Last.fm, MovieLens). Notably, on the highly sparse **Book-Crossing** (99.8% sparsity) and **Taobao** (99.98% sparsity), SURE produces compact sets despite the extreme data sparsity.
>
> ---
>
> ### Q3. The maximum recommended set size was fixed at 25... If this constraint were removed... still maintain compactness?
>
> > Yes, **SURE’s set size distribution still maintains compactness when this constraint is removed**. The compactness is an **inherent property** of the dynamic DAUO thresholding rule (Eq.~15), not a result of an external cap. The cap is included only to reflect standard screen and latency constraints in recommendations.
> >
> > Our experiments in **Section 6.2.2 (Table 2)** **demonstrate** this. That evaluation, which tests the core conformal property (coverage & size), was run **with this $K=25$ constraint removed**. As shown, SURE converges to stable, compact sizes (e.g., 42 items on Book-Crossing) rather than expanding indefinitely, purely via its risk-control mechanism. **This proves that the compactness is a property of SURE's risk control (Eq.15) and is independent of any predefined maximum size.**
>
> ---
>
> ### Q4. If multiple ensemble models collectively exhibit selection bias, can the method still provide accurate preference-based recommendations?
>
> > SURE is robust to selection bias via two mechanisms:
> >
> > 1. **Independent Recalibration:** Each expert is calibrated on its *own* loss. If multiple experts differ from the target risk (bias), their thresholds are adjusted individually (Eq. 15) to restore validity.
> > 2. **Change-Point Reset:** If a correlated bias results in a distribution shift, the $d^{pref}$ metric triggers a segment reset, clearing historical bias and re-initializing weights.
>
> > We have also briefly described how SURE helps in other RS challenges in **Discussion A.9 Appendix**.
>
> ---
>
> ### Q5. Could you incorporate 1–2 relevant works published in 2025 into the experiments to enhance the persuasiveness of the experimental results?
>
> > As discussed in our response to Weakness~1, the recent 2025 adaptive CP papers do not introduce adaptation mechanisms outside the three approaches already covered in our experiments: fixed global thresholds (split CP), sliding/weighted windows (EnbPI), and online quantile updates (online CP). These works operate on scalar residuals and do not support set-valued top-$N$ recommendation problem, user-level shifts, or utility metrics.
> >
> > Adapting them to our setting would require redefining nonconformity scores over interactions and re-deriving coverage guarantees, which constitutes new methodological development rather than a baseline comparison. For completeness, we have cited the most relevant 2025 works in the revised version.
> >
>
>
>
> ### **References**
>
> [1] Shuxin Liang, Yihan Xiao, Linglong Kong, and Wenlu Tang. 2025. Adaptive Conformal Prediction Intervals for Invariant Learning. KDD ’25.
> [2] Junxi Wu, Dongjian Hu, Yajie Bao, Shu-Tao Xia, and Changliang Zou. 2025. Error-quantified Conformal Inference for Time Series. ICLR ’25.
> [3] Alexandros Nanopoulos and Krisztian Buza. 2025. Conformal prediction for out-of-distribution time-series classification. *Applied Intelligence* 55.

---

### Author Response · Authors · 2025-11-24

## Global Response

>We thank the reviewers for distinguishing **SURE** as a significant contribution to the emerging field of **statistically guaranteed recommender systems**. We value the consensus that our work **"fills the research gap"** of lacking strict performance guarantees, supported by **"rigorous theoretical proofs"** and **"comprehensive experimental design"**. We appreciate Reviewers LsNp and mELN for recognizing the **"theoretical guarantee"** and **"practicality"** of our problem formulation. Furthermore, we value Reviewer xUcd for acknowledging the **"clear motivation"** and **"model-agnostic" design**. Before addressing specific questions, we summarize our responses to key concerns:

---

### 1. The "Ensemble" Misconception (Addressing JYLn, mELN)

>On analogies to standard Ensemble Learning or Hedge, we respectfully clarify that this **conflates two distinct mathematical objectives**:
>- **Standard Ensembles [1,2,3]** optimize for *point-prediction accuracy* (minimizing error). They cannot provide statistical coverage guarantees.
>- **SURE** optimizes for **Statistical Validity** (Risk ≤ $\alpha$) and **Set Compactness**.  We utilize Hedge not to minimize error, but with a **novel objective function** (Eq. 9) that penalizes **cumulative set size**.
>- **Key Distinction:**  This reformulation is required to prevent the validity of a conformal predictor from degenerating into impractically large output sets, a specific failure mode of Conformal Prediction that standard ensembles do not address.

---

### 2. Empirical Proof of "Shift" Necessity (Addressing LsNp)

> Whether preference shifts are empirically significant in the chosen datasets,  our results provide **strong empirical evidence** that they are:
>- **The Failure of Baselines:**  If shifts were not real, the standard Split Conformal baseline (calibrated on past data) would maintain the target 90% coverage (1 − α = 0.90).  In our experiments (Table 2), coverage collapsed to **0.82–0.83**.
>- **The SURE Solution:** : This **~7% validity gap** is empirical evidence of distribution shift between calibration and inference. It is not a *'marginal' loss*; in Conformal Prediction, it is a **binary failure of the coverage guarantee**. SURE is necessary *precisely because* it bridges this gap (restoring >90% coverage), validating that static calibration is insufficient for reliable RecSys.

---

### 3. Orthogonality to SOTA (Addressing JYLn, LsNp, xUcd)

>We clarify that SURE is a **model-agnostic calibration layer**, not a competitor to backbone architectures.
>- **Generative & Uncertainty Models:**  Reviewers suggested comparisons to generative (e.g., TIGER [4], HSTU [5] ) or uncertainty-aware models (e.g., ContrastVAE [6]).  In our rebuttal, we applied SURE on top of **HSTU [LsNp]** and **ACVAE [xUcd]**. In both cases, SURE consistently restored statistical coverage guarantees without requiring retraining of the backbone, proving they are **complementary**.
>- **Adaptive CP (2025 works):**  As noted in our response to JYLn, recent scalar regression methods (for eg. [4,5,6] are **task-incompatible** with Top-\(N\) ranking utility (Recall/NDCG) and cannot be applied without the specific reformulation developed in SURE. We have cited them and added them to the Discussion.

---

>We have **highlighted all major revisions** in the updated version in $\color{red}{\text{red}}$.


### References

> [1] Freund, Y., & Schapire, R. E. (1997). A decision-theoretic generalization of on-line learning and an application to boosting. *Journal of Computer and System Sciences*, 55(1), 119–139.
> [2] Cesa-Bianchi, N., & Lugosi, G. (2006). *Prediction, Learning, and Games*. Cambridge University Press.
> [3] Fan, Z., Yu, Z., Yang, K., Chen, W., Liu, X., Li, G., Yang, X., & Chen, C. L. P. (2025). Diverse Models, United Goal: A Comprehensive Survey of Ensemble Learning. *CAAI Transactions on Intelligence Technology*, 10(4), 959–982.
> [4] Rajput, Shashank; et al. Recommender Systems with Generative Retrieval. NeurIPS 2023.
> [5] Deng, Jiaxin; et al. OneRec: Unifying Retrieve and Rank with Generative Recommender and Iterative Preference Alignment. arXiv:2502.18965 (2025).
> [6] Zhe Xie et al. Adversarial and Contrastive Variational Autoencoder for Sequential Recommendation. WWW 2021.
> [7] Shuxin Liang, Yihan Xiao, Linglong Kong, and Wenlu Tang. 2025. Adaptive Conformal Prediction Intervals for Invariant Learning. KDD ’25.
> [8] Junxi Wu, Dongjian Hu, Yajie Bao, Shu-Tao Xia, and Changliang Zou. 2025. Error-quantified Conformal Inference for Time Series. ICLR ’25.
> [9] Alexandros Nanopoulos and Krisztian Buza. 2025. Conformal prediction for out-of-distribution time-series classification. *Applied Intelligence* 55.

---

### Note · Authors · 2026-02-07

I have read and agree with the venue's withdrawal policy on behalf of myself and my co-authors.

---

### Meta-Review · Area_Chair_KGuQ · 2025-12-19

**Summary:**

This paper proposes a model-agnostic framework for sequential recommender systems in the presence of user preference shifts.

Strengths:
(1) An interesting problem (i.e., dynamic shifts in user preferences) in recommender system.
(2) Rigorous theoretic analysis.

weaknesses:
(1) Lack of empirical validation of the preference shift.
(2) Missing recent, stronger baselines.
(3) Limited algorithmic contributions, beyond integrating the existing components.

**Reviewer Concerns:**

(1) The "Ensemble" Misconception, and (2) the orthogonality with sota are addressed.

**Reviewer Scores:**

For all four reviewers: unlike to change the scores.

---

### Decision · Program_Chairs · 2026-01-26

Reject